# The non-ELR CXC chemokine encoded by human cytomegalovirus UL146 genotype 5 contains a C-terminal β-hairpin and induces neutrophil migration as a selective CXCR2 agonist

Christian Berg[1,2], Michael J. Wedemeyer[3], Motiejus Melynis[1], Roman R. Schlimgen[3], Lasse H. Hansen[4,5], Jon Våbenø[6], Francis C. Peterson[3], Brian F. Volkman[3], Mette M. Rosenkilde[1]*, Hans R. Lüttichau[1,2]*

**1** Laboratory for Molecular Pharmacology, Department of Biomedical Sciences, Panum Institute, University of Copenhagen, Copenhagen, Denmark, **2** Unit for Infectious Diseases, Department of Medicine, Herlev-Gentofte Hospital, University of Copenhagen, Herlev, Denmark, **3** Department of Biochemistry, Medical College of Wisconsin, Milwaukee, Wisconsin, USA, **4** Department of Clinical Biochemistry, Rigshospitalet, University of Copenhagen, Copenhagen, Denmark, **5** Copenhagen Center for Glycomics, Department of Cellular and Molecular Medicine, Faculty of Health Sciences, University of Copenhagen, Copenhagen N, Denmark, **6** Helgeland Hospital Trust, Sandnessjøen, Norway

\* rosenkilde@sund.ku.dk (MMR); Hans.Rudolf.von.Luttichau@regionh.dk (HRL)

## Abstract

Human cytomegalovirus (HCMV) is a major pathogen in immunocompromised patients. The UL146 gene exists as 14 diverse genotypes among clinical isolates, which encode 14 different CXC chemokines. One genotype (vCXCL1$_{GT1}$) is a known agonist for CXCR1 and CXCR2, while two others (vCXCL1$_{GT5}$ and vCXCL1$_{GT6}$) lack the ELR motif considered crucial for CXCR1 and CXCR2 binding, thus suggesting another receptor targeting profile. To determine the receptor target for vCXCL1$_{GT5}$, the chemokine was probed in a G protein signaling assay on all 18 classical human chemokine receptors, where CXCR2 was the only receptor being activated. In addition, vCXCL1$_{GT5}$ recruited β-arrestin in a BRET-based assay and induced migration in a chemotaxis assay through CXCR2, but not CXCR1. In contrast, vCXCL1$_{GT1}$ stimulated G protein signaling, recruited β-arrestin and induced migration through both CXCR1 and CXCR2. Both vCXCL1$_{GT1}$ and vCXCL1$_{GT5}$ induced equally potent and efficacious migration of neutrophils, and ELR vCXCL1$_{GT4}$ and non-ELR vCXCL1$_{GT6}$ activated only CXCR2. In contrast to most human chemokines, the 14 UL146 genotypes have remarkably long C-termini. Comparative modeling using Rosetta showed that each genotype could adopt the classic chemokine core structure, and predicted that the extended C-terminal tail of several genotypes (including vCXCL1$_{GT1}$, vCXCL1$_{GT4}$, vCXCL1$_{GT5}$, and vCXCL1$_{GT6}$) forms a novel β-hairpin not found in human chemokines. Secondary NMR shift and TALOS+ analysis of vCXCL1$_{GT1}$ supported the existence of two stable β-strands. C-terminal deletion of vCXCL1$_{GT1}$ resulted in a non-functional protein and in a shift to solvent exposure for tryptophan residues likely due to destabilization of the chemokine fold. The results demonstrate that non-ELR chemokines can activate CXCR2 and

**Data Availability Statement:** All relevant data are within the manuscript and its Supporting Information files.

**Funding:** C.B. was supported by the Department of NeuroScience and Pharmacology (120-0016/14-3000 ) – now Department of Biomedical Sciences (https://bmi.ku.dk/) - Faculty of Health and Medical Sciences, University of Copenhagen (Denmark); the Faculty of Health and Medical Sciences, University of Copenhagen (Denmark) (A5151); the Department of Medicine, Herlev-Gentofte Hospital, Denmark (https://www.herlevhospital.dk/afdelinger-og-klinikker/Afdeling-for-Medicinske-Sygdomme/Sider/default.aspx); Herlev and Gentofte Hospital's Research Council (https://www.herlevhospital.dk/english/Sider/InformaAbout-Research-at-Herlev-and-Gentofte-Hospital.aspx); Carl and Ellen Hertz Grant (7179-2), Christian Larsen and Judge Ellen Larsen's Grant (5011780), Dagmar Marshall's Foundation (https://www.marshallsfond.dk/); the Hartmann Foundation (A31960) and the A.P. Møller Foundation for the Promotion of the Medical Science (17-L-0446). M. M.R was supported by the European Research Council: VIREX (Grant agreement 682549, Call ERC-2105-CoG). C.B. and M.M.R. were supported by a PreSeed grant from the Novo Nordic Foundation: NNFDOC-1367246185-166. M.J.W., R.R.S., F.C.P. and B.F.V. were supported by NIH grant R37 AI058072. The funders had no role in study design, data collection and analysis, decision to publish, or preparation of the manuscript.

**Competing interests:** I have read the journal's policy and the authors of this manuscript have the following competing interests: B.F.V. and F.C.P. have ownership interests in Protein Foundry and XLock Biosciences.

**Abbreviations:** HCMV, human Cytomegalovirus; CXCR, C-X-C chemokine receptor; CCR, C-C chemokine receptor; CXCL, C-X-C chemokine ligand; CCL, C-C chemokine ligand; vCXCL1, viral C-X-C chemokine ligand 1; JCAT, JAVA Codon Adaptation Tool; PBN, peripheral blood neutrophil; GRK, G protein receptor kinase; IP, inositol phosphate; BRET, bioluminescence resonance energy transfer; nanoDSF, nano differential scanning fluorimetry; NMR, nuclear magnetic resonance; PDB, Protein Data Bank; TALOS+, torsion angle likelihood obtained from shift and sequence similarity (version +).

suggest that the UL146 chemokines have unique C-terminal structures that stabilize the chemokine fold. Increased knowledge of the structure and interaction partners of the chemokine variants encoded by UL146 is key to understanding why circulating HCMV strains sustain 14 stable genotypes.

## Author summary

Human cytomegalovirus (HCMV) is a prevalent herpesvirus infecting an estimated 60% of the human population worldwide. It is commonly transmitted during early childhood and leads to life-long latency, where viral reactivation can cause severe complications in case of host immune suppression. Furthermore, HCMV is the leading cause of congenital infections. Circulating HCMV strains exhibit great genetic diversity unusual for DNA viruses. One of its most diverse genes is UL146, which encodes a chemokine that facilitates viral dissemination by exploiting the human immune system through mimicry of key immunity components. In this study, we investigate how the diversity of UL146 affects its signaling and structural properties to understand why its genetic diversity is maintained across human populations. We find that certain genotypes that lack key structural domains present in the human homologs nonetheless exert similar functions in the virus-host relationship. Furthermore, many of the UL146 genotypes contain novel structural elements critical for correct protein folding and with the potential to provide HCMV with additional immune modulatory and evasive features. Together, our data highlight a considerable degree of host-adaptation by HCMV and propose novel structural interactions with implications for the virus-host interplay.

## Introduction

Human cytomegalovirus (HCMV) is a highly prevalent human herpesvirus that causes severe disease in immunocompromised patients. Both primary infection and reactivation of latent infection is a significant concern among HIV/AIDS patients, organ transplant recipients, in chemotherapy, and during pregnancy due to the risk of mother-to-child transmission, as it can lead to disability and death [1].

The 235 kb genome of HCMV encodes a large number of immune modulators that are used for immune evasion, viral dissemination, and latency. A common feature for herpesviruses is that a large part of their immunomodulatory genes are somehow connected to the chemokine system, as these viruses either encode chemokine receptors or chemokine ligands used to manipulate the host immune system, or they modulate the expression of host-encoded chemokine receptors [2,3]. Previous studies have observed a high degree of genetic diversity among different clinical HCMV isolates, including disruptive mutations to many immunomodulatory genes [4,5]. One of the most diverse genes, UL146, encodes the chemokine vCXCL1 [6] of which 14 distinct genotypes have been identified from clinical isolates [7,8].

We have previously demonstrated that UL146 genotype 1 (vCXCL1$_{GT1}$, corresponding to the UL146 encoded by the Toledo strain) induces chemotaxis through its agonistic activity on G protein signaling of the neutrophil chemokine receptors CXCR1 and CXCR2 [9]. The attraction of neutrophils [10,11] to infection sites has been shown to facilitate viral dissemination and has established vCXCL1 as a virulence factor in a mouse model [12]. CXCR2 has been suggested to be the main receptor for vCXCL1-induced migration as CXCR1$^+$ CXCR2$^-$

natural killer (NK) cells are co-attracted to a lesser extent than neutrophils [11]. Differences of variable importance in CXCR1/CXCR2 binding affinity, signaling potency, and chemotactic properties have been observed for several UL146 genotypes [10], but, apart from genotype 1 [9], the 13 remaining genotypes are yet to be characterized for functional activity against the remaining 17 chemokine receptors including CXCR1. All endogenous chemokines for CXCR1 and CXCR2 contain an ELR (Glu-Leu-Arg) motif in their N-terminus [13] and any change to the ELR motif in these chemokines eliminates their signaling through CXCR1 and CXCR2 [14]. Furthermore, introduction of an ELR motif in a non-ELR chemokine allows it to bind to CXCR1 and CXCR2 [15]. Thus, the ELR motif has been considered a molecular signature for CXCR1/2-targeting chemokines. Interestingly, UL146 genotypes 5 and 6 lack this motif (Fig 1), which suggests that these two UL146 variants target different chemokine receptors, which would imply an altered biological effect of the gene and, potentially, a change in pathogenicity for HCMV strains encoding UL146 genotype 5 and 6. Furthermore, no consensus on the signal peptide cleavage site exists for the non-ELR genotypes with published sequences defaulting to the same site as the other genotypes [10] while the signal peptide prediction server SignalP 4.1 [16] suggests a cleavage site that results in a longer N-terminus. These substantial differences in N-terminus length, ELR motifs, and general sequence variability are further complicated by differences in predicted glycosylation patterns (Figs 1 and 2A) [17,18]. Additionally, these HCMV-encoded chemokines have an extended C-terminus, which is only seen in seven of the nearly 50 human chemokines (XCL1, CCL16, CCL21, CCL25, CCL28, CXCL9 and CXCL12γ). Experimental structures of CCL25 and CXCL9 have not been reported, but NMR studies of XCL1 [19], CCL21 [20], CCL28 [21], and CXCL12γ [22] have demonstrated unstructured C-termini, while the distal C-terminus is not represented in an X-ray crystal structure of CCL16 (PDB ID: 5LTL).

In this study, we characterize the receptor targeting profile of the non-ELR chemokine vCXCL1$_{GT5}$ and analyze its receptor activity in parallel with vCXCL1$_{GT1}$ in G protein signaling, β-arrestin recruitment, and chemotaxis assays, and we compare their signaling capabilities to those of another ELR (vCXCL1$_{GT4}$) and non-ELR (vCXCL1$_{GT6}$) genotype. Furthermore, we investigate the effect of N- and C-terminal chemokine truncations on receptor activation of selected vCXCL1 genotypes. Finally, we generate *in silico* structural models for all 14 genotypes using Rosetta to gain insight into the tertiary structures of these viral chemokines, and we compare the modelling to *in vitro* data of the vCXCL1$_{GT1}$ tail structure generated by NMR and TALOS+ analysis.

## Results

### The mature vCXCL1$_{GT5}$ is a 97 amino acid glycosylated chemokine with an extended N-terminus and a predicted novel C-terminal β-hairpin

Viral gene codon usage (e.g. for HIV and herpesviruses) can be unusual, resulting in poor or no expression of viral proteins unless a viral expression regulator is co-expressed [26]. Thus, we determined the codon availability indexes (CAI) of UL146 genotypes 1 and 5 genes using the JCAT algorithm [27], and both had a very low CAI of 0.10 indicating a highly unusual codon usage and low expression. To ensure sufficient expression of vCXCL1$_{GT1}$ and vCXCL1$_{GT5}$, the UL146 genes were codon-optimized (resulting in CAI values of 0.69 and 0.64, respectively), synthesized, and inserted in a eukaryotic expression vector.

To ensure that the mature chemokines corresponded to those naturally secreted from HCMV-infected cells, vCXCL1$_{GT1}$ and vCXCL1$_{GT5}$ were expressed in eukaryotic COS-7 cells. The chemokines were collected in conditioned serum-free medium and purified by cation-exchange followed by HPLC purification, which gave two major unique peaks eluting at 31%

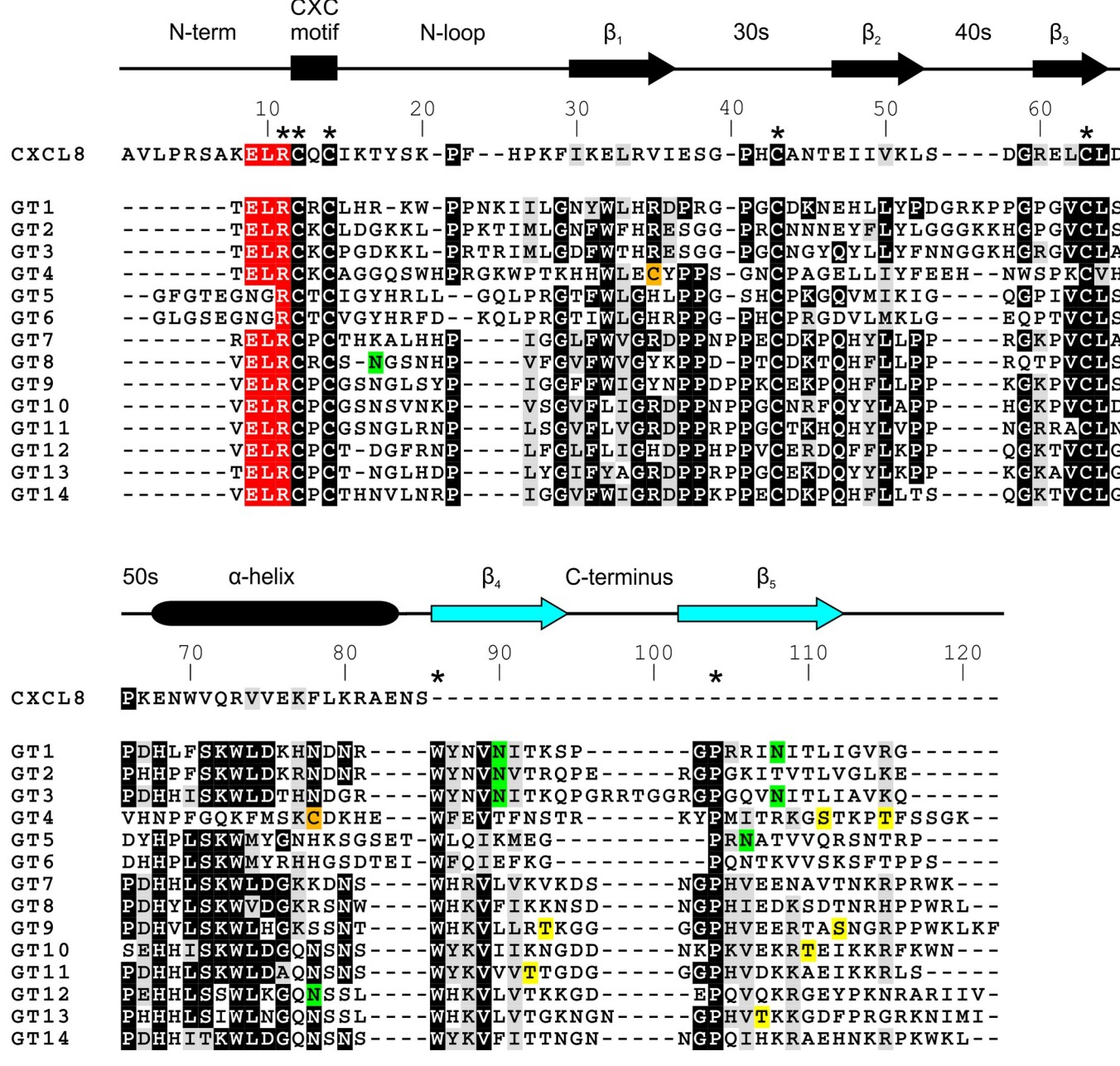

**Fig 1. Sequence alignments of the endogenous human chemokine CXCL8 and the 14 highly divergent genotypes of the HCMV-encoded UL146 chemokine.** Alignment generated with ClustalW 2.1 and adapted to illustrate differences in secondary structure. White on black illustrates identical residues, and black on grey similar residues. Important differences are color coded. The ELR motif is marked white on red and residues 100% conserved in UL146 are marked with a star (*). Genotypes 5 and 6 are shown with the longer N-terminus predicted by the SignalP 4.1 server [16]. Predicted glycosylation sites are marked black on green (N-glycosylation) and black on yellow (O-glycosylation) based on the NetNGlyc 1.0 [17] and NetOGlyc 4.0 [18] server predictions (threshold = 0.5). The two additional cysteine residues in the vCXCL1$_{GT4}$ sequence (suggesting a third disulfide bond) are marked black on orange. The secondary structures of the chemokines are shown above the sequences as determined by NMR and X-ray structures for CXCL8 [23–25] and determined by Rosetta modelling for vCXCL1$_{GT1–GT14}$. The proposed fourth and fifth β-strands unique to several vCXCL1 genotypes of the UL146 encoded chemokines have been marked by light blue (for the exact location of the α-helix and the β-strands for a particular UL146 genotype see S1 Fig).

and 37% acetonitrile respectively (Fig 2B) that were not present in the medium of cells transfected with the empty expression vector. MALDI-TOF analysis of the corresponding isolated fractions revealed pure samples with three peaks at 12188–12791 Da for vCXCL1$_{GT5}$ (Fig 2C).

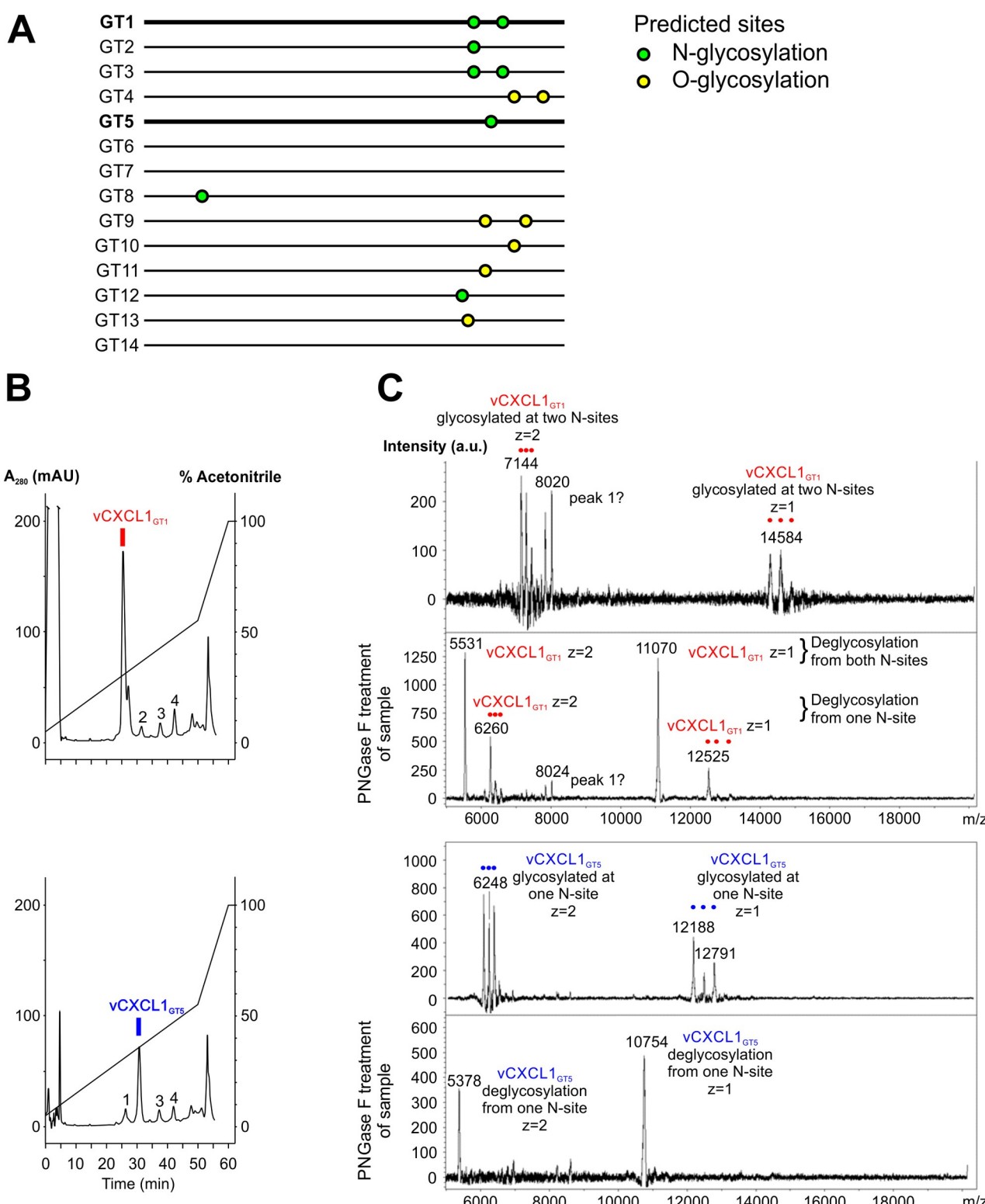

**Fig 2. Purification and mass spectrometry analysis of recombinant vCXCL1$_{GT1}$ and vCXCL1$_{GT5}$ expressed in COS-7 cells. (A)** Cartoon representation of predicted glycosylation sites in the 14 genotypic vCXCL1 proteins **(B)** HPLC elution profiles of conditioned media from eukaryotic cells transfected with UL146$_{GT1}$ (top panel) and UL146$_{GT5}$ (bottom panel). Minor background peaks are marked numerically. mAU; milliabsorbance units. **(C)** MALDI-TOF mass spectrometry analysis of untreated (top chromatogram) and PNGase F treated (bottom chromatogram) samples of vCXCL1$_{GT1}$ and vCXCL1$_{GT5}$ (red and blue peaks isolated in panel B).

The predicted mass of the vCXCL1$_{GT5}$ was 10730 Da, but as the chemokine was likely N-glycosylated at position 86 (Fig 1), we treated the sample with PNGase F, which yielded a single peak at 10754 Da (Fig 2C), corresponding to the predicted mass of the full 97-residue vCXCL1$_{GT5}$ plus a sodium ion (Na$^+$ of 23 Da, a common adduct). Trypsinization identified the chemokine with 37.1% coverage (S2 Fig).

These experiments show that the signal peptide cleavage site for vCXCL1$_{GT5}$ is positioned between residues 19 and 20 of the precursor protein, resulting in an extended N-terminus (nine residues) compared to the ELR vCXCL1 chemokines (four residues) (Fig 1).

For vCXCL1$_{GT1}$, two N-glycosylation sites at positions 76 and 87 were predicted (Fig 1) and MALDI-TOF showed three peaks at 14290–14889 Da (Fig 2C). In accordance with the predicted signal peptide cleavage site between residues 22 and 23 of the precursor protein yielding a mature chemokine mass of 11073 Da, PNGase F-treatment resulted in a single peak of 11070 Da corresponding to a short N-terminus of four residues (Fig 1). The mass loss upon deglycosylation was approximately twice as large for vCXCL1$_{GT1}$ (3220–3819 Da) as for vCXCL1$_{GT5}$ (1434–2037 Da), which fits with the number of predicted N-glycosylation sites in the two proteins.

Identifying the mature forms of vCXCL1$_{GT1}$ and vCXCL1$_{GT5}$ enabled an increased chemokine production by expression of the wild-type (1–97) genotype 5 UL146 in *E. coli* as well as the previously characterized genotype 1 UL146 for reference (Fig 1). As previous studies have shown that N-terminal truncation of the endogenous CXCR1 and CXCR2 ligands CXCL1 and CXCL8 can lead to increased activity [28–30], we also expressed two N-terminally truncated versions of vCXCL1$_{GT5}$, vCXCL1$_{GT5\ (4–97)}$ and vCXCL1$_{GT5\ (6–97)}$, corresponding to an alternate predicted signal peptide cleavage site (4–97) and the cleavage site of the regular ELR vCXCL1s (6–97) (Fig 1). Furthermore, the other non-ELR chemokine (vCXCL1$_{GT6}$) and an additional ELR chemokine (vCXCL1$_{GT4}$) were also expressed in order to probe for differences in receptor activation within both non-ELR and ELR groups of the vCXCL1 chemokines. Finally, a tailless version of vCXCL1$_{GT1}$ was expressed to examine the function of the extended C-terminus. Recombinant proteins were generated by two different protocols. In short, inclusion bodies were isolated, solubilized, and the chemokines underwent refolding and purification by FPLC cation-exchange chromatography followed by either size-exclusion chromatography or HPLC. Peptide masses were confirmed by mass-spectrometry. These recombinant UL146 chemokines were used for all experiments presented in this paper.

To gain structural insight into vCXCL1$_{GT1}$ and vCXCL1$_{GT5}$, we used Rosetta to generate comparative models of both genotypes based on reported structures of CXCL2 (PDB ID: 3N52), CXCL5 (PDB ID: 2MGS), and CXCL8 (PDB ID: 5WDZ). These models were assessed by Rosetta total energy and the energetically favored top models (i.e. lowest Rosetta total score) were visually inspected. The top models for both vCXCL1$_{GT1}$ and vCXCL1$_{GT5}$ adopt the classic chemokine fold, with an unstructured N-terminus (Fig 3A and 3D) leading into an antiparallel 3-stranded β-sheet (Fig 3B and 3E) followed by an α-helix. Although some human chemokines have additional unstructured residues at the C-terminus, most terminate shortly after the α-helix. Both vCXCL1$_{GT1}$ and vCXCL1$_{GT5}$ contain an extended C-terminal tail of approximately 25 residues, which form a β-hairpin motif in all top models, although the spatial position of this motif varies (Figs 3C and 3F, and S1).

Next, we generated models for the remaining vCXCL1 genotypes, each of which adopted the classical chemokine structure (S3A Fig). Of the 14 genotypes, three (vCXCL1$_{GT3}$, vCXCL1$_{GT13}$, and vCXCL1$_{GT14}$) showed a loop-turn-loop conformation for the extended C-terminus, while the other 11 all displayed a C-terminal β-hairpin (S1 and S3B Figs).

To assess the reliability of the molecular modeling, we used Nuclear Magnetic Resonance (NMR) to assign the backbone chemical shifts of vCXCL1$_{GT1}$. A heteronuclear single-quantum coherence (HSQC) spectrum showed well-dispersed peaks indicative of a folded protein,

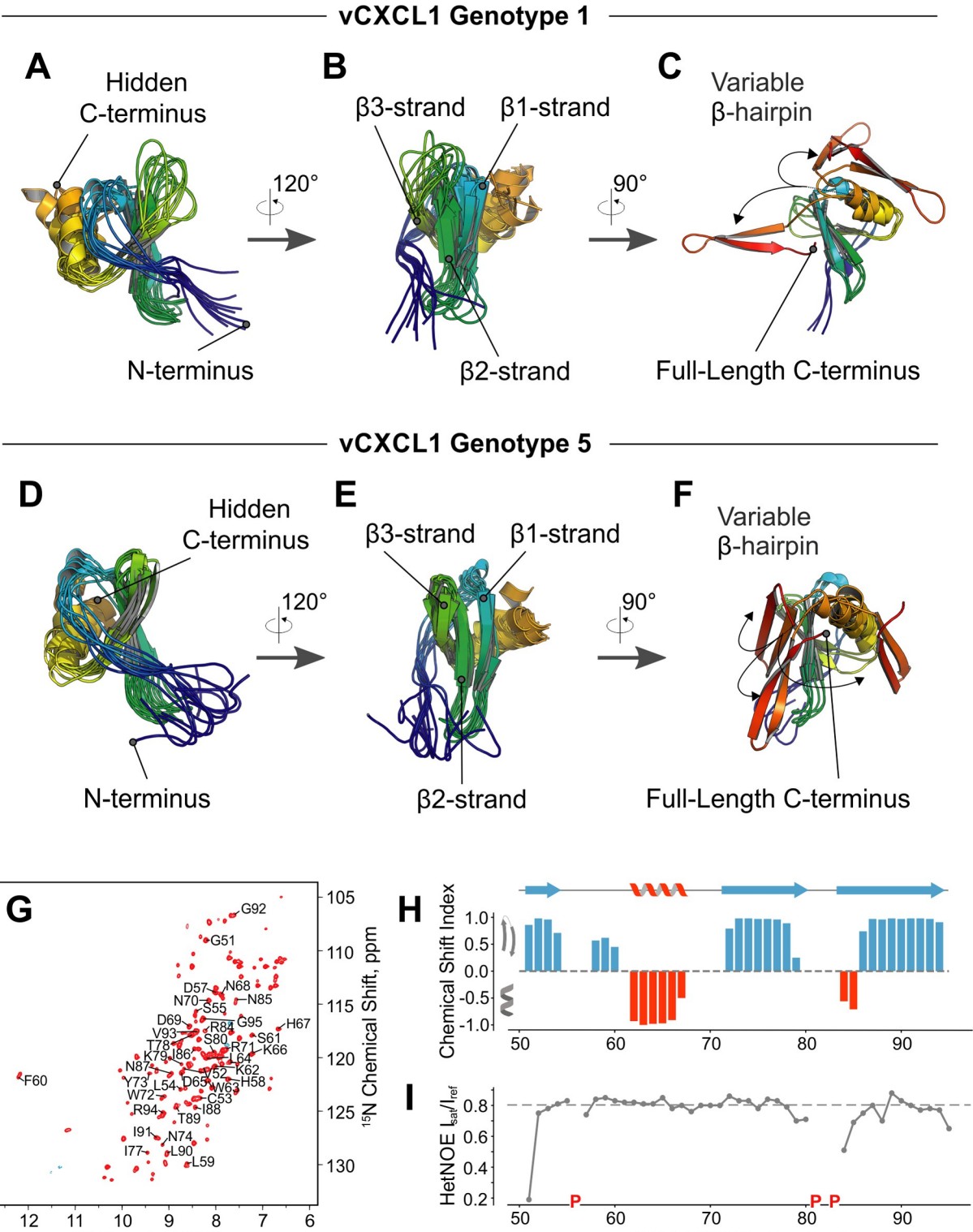

**Fig 3. Structural analysis of vCXCL1 GT1 and vCXCL1 GT5 via molecular modeling and Nuclear Magnetic Resonance (NMR) Spectroscopy.** Cartoon representations of the top ten models (lowest Rosetta total score) of vCXCL1GT1 are overlaid and viewed from the left **(A)** and front **(B)**. Each polypeptide chain is rainbow colored from N- (blue) to C-terminus (red). The extended C-terminus is hidden to highlight the convergence in the modeling (core Cα-RMSD: 3.9Å) and formation of the expected C-terminal α-helix. **(C)** Right-side view of three of the top vCXCL1GT1 models with the wild-type C-terminus, which forms a β-hairpin in all cases, but with no consensus location. **(D)**

The ten top models of vCXCL1$_{GT5}$ are shown overlaid and viewed from the left (core Cα-RMSD: 1.8Å) and front (**E**). (**F**) Right-side view of three of the top vCXCL1$_{GT5}$ models with the wild-type C-terminus, which forms a β-hairpin in all cases, but with no consensus position. (**G**) NMR $^1$H-$^{15}$N HSQC spectrum of the C-terminus of vCXCL1$_{GT1}$ (residues 51–95). Backbone NH assignments are indicated by the one letter amino acid code and residue number. (**H**) Predicted secondary structure for vCXCL1$_{GT1}$ based on secondary chemical shifts for residues 51–95 using TALOS+. The modeled secondary structure of vCXCL1$_{GT1}$ using RosettaCM is displayed above. (**I**) Plot of {$^1$H}-$^{15}$N Heteronuclear NOE values reflecting the picosecond-nanosecond motions of the vCXCL1$_{GT1}$ C-terminus. Proline residues are indicated with a red P, a gap at position 82 indicates inconclusive assignment information for this residue.

including the shifts corresponding to the extended C-terminus (Fig 3G). Using TALOS+, we parsed secondary structure elements in the extended vCXCL1$_{GT1}$ C-terminus based on the $^1$H, $^{13}$Cα, $^{13}$Cβ, $^{13}$C′, and $^{15}$N chemical shifts [31]. The TALOS+ analysis predicted β-strands for residues 72–79 and 85–94 with high confidence, matching the consensus location of the β-hairpin in the Rosetta models (Fig 3H). Additionally, high {$^1$H}-$^{15}$N heteronuclear NOE values (>0.6) in the C-terminus suggested that the vCXCL1$_{GT1}$ protein backbone in this region is well ordered (Fig 3I).

## vCXCL1$_{GT5}$ is a selective CXCR2 agonist in G protein signaling

We investigated the receptor target(s) of the three non-ELR vCXCL1$_{GT5}$ chemokine variants by screening for both agonistic and antagonistic activity on G protein signaling on a panel of all 18 classical human chemokine receptors in an IP accumulation assay (Fig 4). The HCMV-encoded chemokine receptor US28$_{\Delta300}$ (a C-terminally truncated US28 with increased signaling properties [32]) was also included. vCXCL1$_{GT5}$ was found to activate CXCR2 at ~60% of the maximal activity of CXCL8 and did not activate CXCR1 or any other chemokine receptor (Fig 4B), whereas the two N-terminally truncated variants (vCXCL1$_{GT5\ (4–97)}$ and vCXCL1$_{GT5\ (6–97)}$) were half as efficacious as the wild-type chemokine on CXCR2 at ~30% (31% and 29%, respectively) the efficacy of the CXCL8 on CXCR2. The results support that the 1–97 variant with the extended N-terminus is the secreted functional form of vCXCL1$_{GT5}$. Pretreatment of the chemokine receptors with 1 μM vCXCL1$_{GT5}$ did not block or inhibit the response of any of the 18 human chemokine receptors or US28$_{\Delta300}$ (Fig 4B), and the same results were seen for the truncated vCXCL1$_{GT5}$ variants. Thus, no antagonistic activity on G protein signaling was observed, and as vCXCL1$_{GT5}$ was unable to activate or inhibit the US28 receptor we found no evidence that vCXCL1$_{GT5}$ participated in an autocrine loop through US28. These experiments demonstrate that vCXCL1$_{GT5}$ is a CXCR2 selective agonist with respect to G protein signaling.

## vCXCL1$_{GT5}$ causes IP accumulation through CXCR2 in a dose-dependent manner

Having established that vCXCL1$_{GT5}$ selectively targets CXCR2, in contrast to vCXCL1$_{GT1}$ that targets both CXCR1 and CXCR2 [9], we decided to investigate the dose/response relationship of G protein activation of the two chemokine receptors using two ELR vCXCL1s (genotype 1 and genotype 4) and the two non-ELR vCXCL1s (genotype 5 and genotype 6) with CXCL8 for reference (Fig 5A–5D). Additionally, the two N-terminally truncated vCXCL1$_{GT5}$ variants were tested (Fig 5E and 5F). On CXCR1, vCXCL1$_{GT1}$ activated G protein signaling with a lower potency than CXCL8, while no activity was seen for vCXCL1$_{GT4}$ (Fig 5A). On CXCR2, vCXCL1$_{GT1}$ and vCXCL1$_{GT4}$ matched the activity of CXCL8 (Fig 5B). The two non-ELR vCXCL1s (vCXCL1$_{GT5}$ and vCXCL1$_{GT6}$) were, like the ELR chemokine vCXCL1$_{GT4}$, unable to activate CXCR1 (Fig 5C), but activated CXCR2 to the same extent although with lower potency than CXCL8 (Fig 5D). The two N-terminally truncated vCXCL1$_{GT5}$ variants showed the same pattern of activation as vCXCL1$_{GT5}$, but did not significantly activate the receptor at concentrations below 1 μM (Fig 5E and 5F).

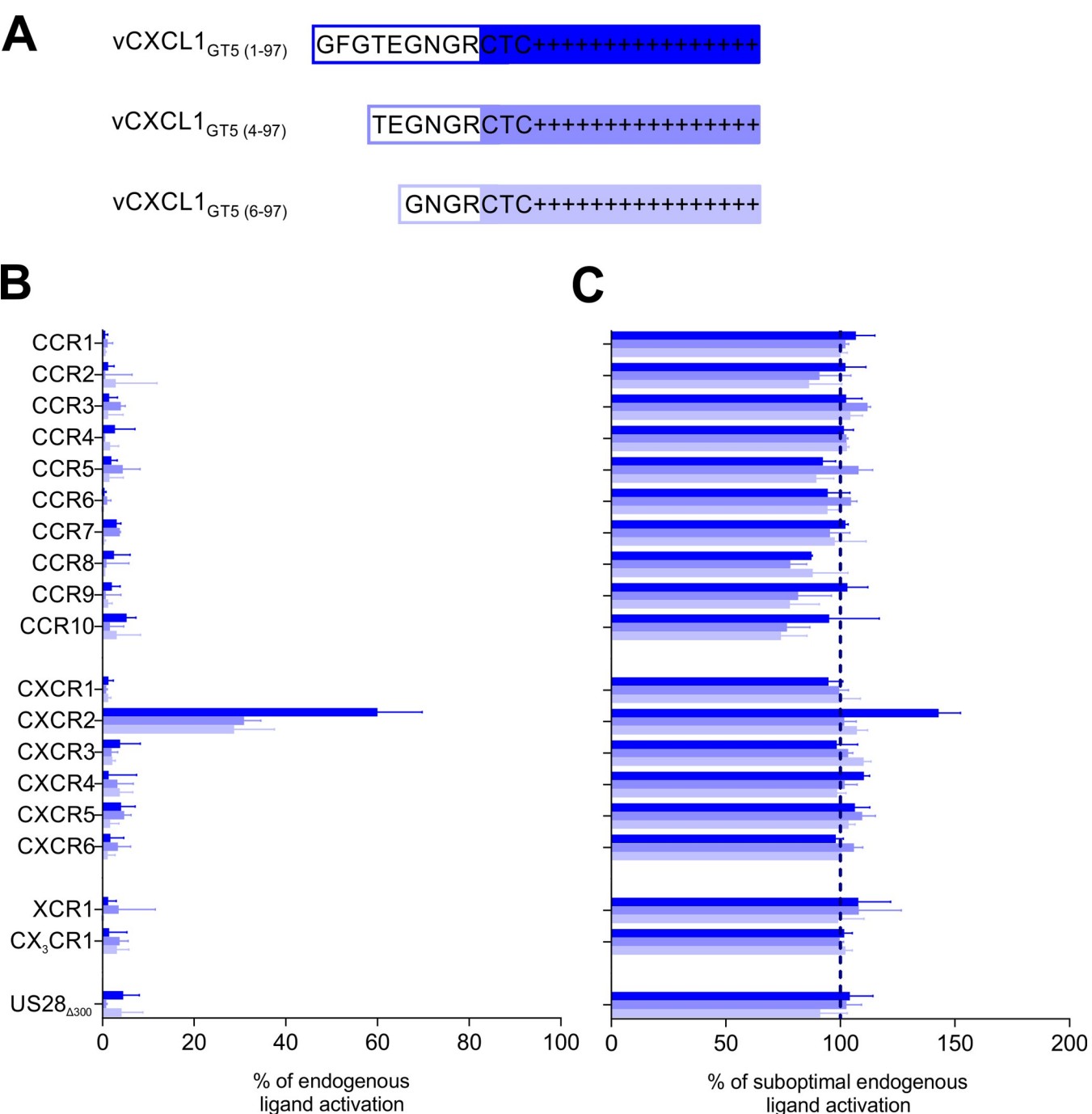

**Fig 4. Screening of non-ELR vCXCL1$_{GT5}$ activity on G protein signaling of an array of all 18 classical human chemokine receptors and the HCMV-encoded US28. (A)** N-terminal sequences of the wild-type and truncated versions of vCXCL1$_{GT5}$. **(B)** Agonism of 1 μM wild-type vCXCL1$_{GT5}$, vCXCL1$_{GT5\ (4-97)}$, and vCXCL1$_{GT5\ (6-97)}$ on chemokine receptor G protein signaling in an IP accumulation assay compared to the response of the endogenous ligand for the receptor. The following endogenous chemokines were used as a positive control (concentrations shown in brackets): CCR1;CCL5 (B: 100 nM, C: 10 nM), CCR2;CCL7 (B: 100 nM, C: 10 nM), CCR3;CCL11 (B: 100 nM, C: 10 nM), CCR4;CCL17 (B: 100 nM, C: 10 nM), CCR5;CCL5 (B: 100 nM, C: 10 nM), CCR6; CCL20 (B: 100 nM, C: 1 nM), CCR7;CCL19 (B: 100 nM, C: 10 nM), CCR8;CCL1 (B: 100 nM, C: 10 nM), CCR9;CCL25 (B: 100 nM, C: 10 nM), CCR10;CCL27 (B: 100 nM, C: 10 nM), CXCR1;CXCL8 (B: 1 μM, C: 10 nM), CXCR2;CXCL8 (B: 1 μM, C: 100 nM), CXCR3;CXCL11 (B: 100 nM, C: 10 nM), CXCR4;CXCL12 (B: 100 nM, C: 10 nM), CXCR5;CXCL13 (B: 100 nM, C: 10 nM), CXCR6;CXCL16 (B: 100 nM, C: 1 nM), XCR1;XCL1 (B: 100 nM, C: 100 nM), CX$_3$CR1; CX$_3$CL1 (B: 100 nM, C: 1 nM), US28$_{\Delta300}$;CX$_3$CL1 (B: 100 nM, C: 1 nM); n = 3–6. All error bars are presented as SEM. **(C)** Antagonism of 1 μM wild-type vCXCL1$_{GT5}$, vCXCL1$_{GT5\ (4-97)}$, and vCXCL1$_{GT5\ (6-97)}$ on chemokine receptor G protein signaling in an IP accumulation assay using submaximal concentrations of the endogenous chemokines described above; n = 3–6. All error bars are presented as SEM.

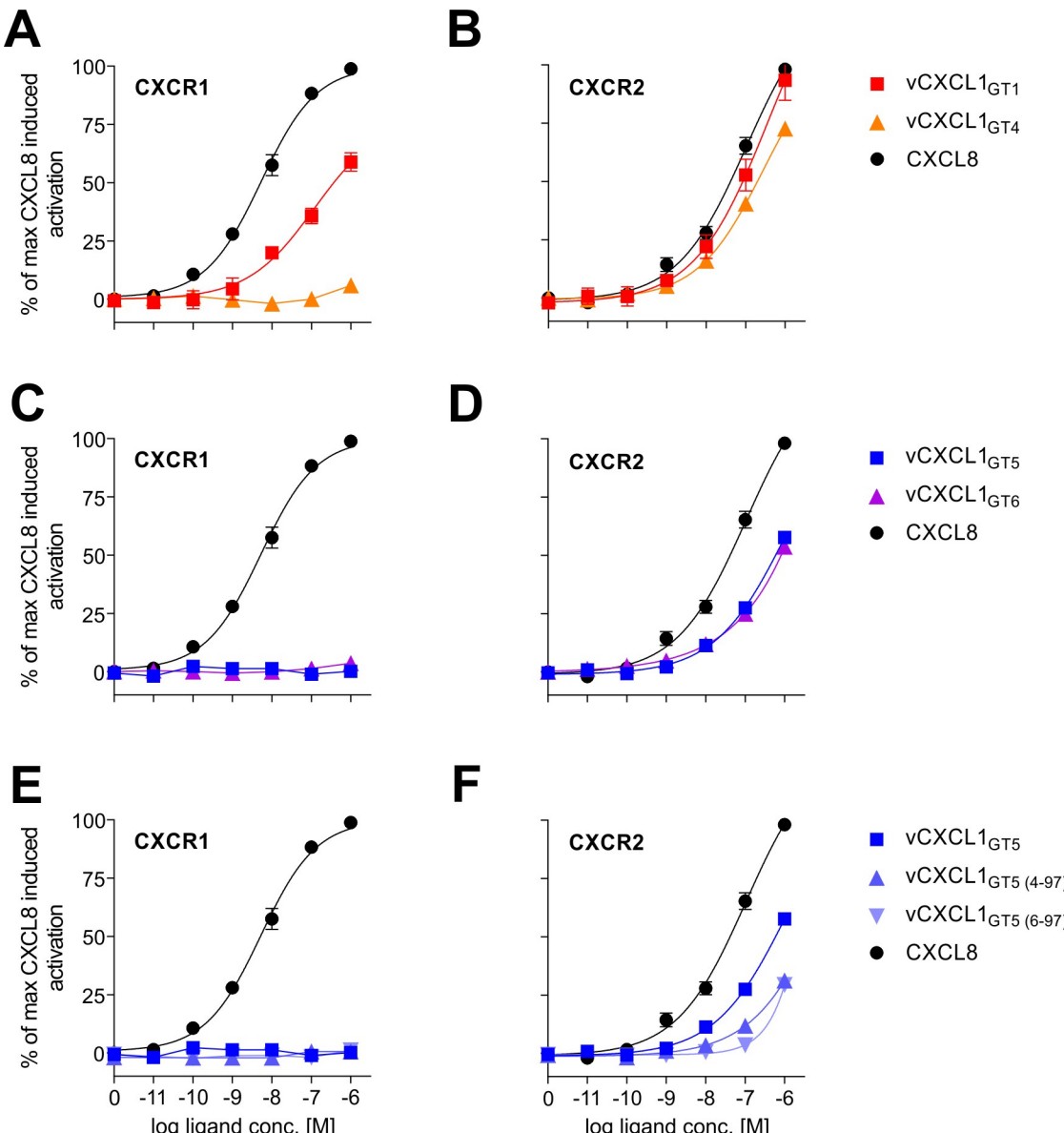

**Fig 5. Dose/response activity of vCXCL1$_{GT1}$, vCXCL1$_{GT4}$, vCXCL1$_{GT6}$, and vCXCL1$_{GT5}$ variants on G protein signaling of CXCR1 and CXCR2.** G protein signaling of CXCR1 and CXCR2 by vCXCL1$_{GT1}$ and vCXCL1$_{GT4}$ **(A+B)**, wild-type vCXCL1$_{GT5}$ and vCXCL1$_{GT6}$ **(C+D)**, and vCXCL1$_{GT5 (4–97)}$, and vCXCL1$_{GT5 (6–97)}$ **(E+F)** in an IP accumulation assay. Error bars presented as SEM. For CXCR1, the number of repeated observations for each sum curve were: CXCL8 n = 10; vCXCL1$_{GT1}$ n = 8; vCXCL1$_{GT4}$ n = 3; wild-type vCXCL1$_{GT5}$ n = 10; vCXCL1$_{GT6}$ n = 3; vCXCL1$_{GT5 (4–97)}$ n = 4; vCXCL1$_{GT5 (6–97)}$ n = 4. For CXCR2, the number of repeated observations for each sum curve were: CXCL8 n = 10; vCXCL1$_{GT1}$ n = 8; vCXCL1$_{GT4}$ n = 3; wild-type vCXCL1$_{GT5}$ n = 10; vCXCL1$_{GT6}$ n = 3; vCXCL1$_{GT5 (4–97)}$ n = 3; vCXCL1$_{GT5 (6–97)}$ n = 3.

From this dose/response investigation, it was clear that the N-terminally truncated vCXCL1$_{GT5}$ variants had severely inhibited signaling properties compared to the wild-type chemokine (Fig 5F), which suggests that the first three residues of the N-terminus, Gly-Phe-Gly (GFG), are crucial to the activity of vCXCL1$_{GT5}$. As expected, the two tested ELR vCXCL1s followed the same pattern of receptor activation as human ELR chemokines in either selectively targeting CXCR2 (vCXCL1$_{GT4}$) or targeting both CXCR1 and CXCR2 (vCXCL1$_{GT1}$).

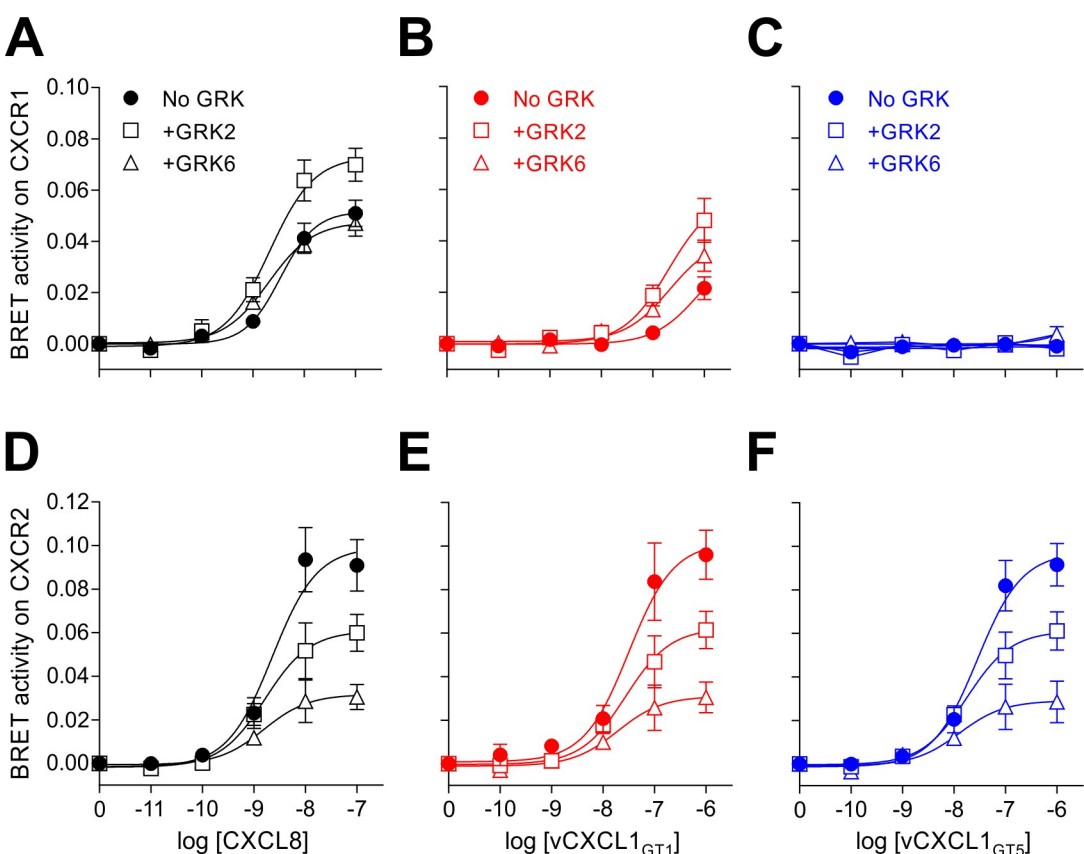

**Fig 6. The effect of vCXCL1$_{GT1}$ and vCXCL1$_{GT5}$ on β-arrestin recruitment through CXCR1 and CXCR2 with/without GRKs.** Agonism of CXCL8 (**A+D**), vCXCL1$_{GT1}$ (**B+E**), and wild-type vCXCL1$_{GT5}$ (**C+F**) on the β-arrestin recruitment pathway of CXCR1 and CXCR2 in a BRET-based assay without GRKs (circle, ●), with GRK2 (square, ■), and with GRK6 (triangle, ▲); n = 3–4. Error bars presented as SEM.

### Both ELR and non-ELR vCXCL1 stimulate β-arrestin recruitment

A previous study has demonstrated the ability of several vCXCL1s to recruit β-arrestin through CXCR2 [10]. In order to investigate the ability of vCXCL1$_{GT1}$ and vCXCL1$_{GT5}$ to recruit β-arrestin by both CXCR1 and CXCR2 they were tested in a BRET-based β-arrestin recruitment assay using CXCL8 as a positive control (Fig 6). We also tested how G protein-coupled receptor kinase (GRK) phosphorylation affected β-arrestin recruitment by co-transfecting with GRK2 and GRK6. The results mirrored those from the G protein signaling assay, as vCXCL1$_{GT1}$ was able to stimulate β-arrestin recruitment on CXCR1 (Fig 6B) at >100-fold lower potency than CXCL8 (Fig 6A) while vCXCL1$_{GT5}$ did not (Fig 6C). For CXCR2, both vCXCL1$_{GT1}$ and vCXCL1$_{GT5}$ stimulated β-arrestin recruitment at comparable efficacies to CXCL8 but with ~10-fold lower potencies (Fig 6D–6F).

Addition of GRK2 increased the activity of CXCL8 and vCXCL1$_{GT1}$ on CXCR1 slightly (Fig 6A and 6B), while GRK6 co-transfection did not have a significant effect, suggesting that GRK2 is involved in the phosphorylation of CXCR1 upon CXCL8 and vCXCL1$_{GT1}$-induced activation. The opposite was observed for CXCR2, as GRK2 reduced the activity of all three ligands considerably and GRK6 did so to an even greater extent (Fig 6D–6F). This GRK dependency agrees with previous observations for CXCL8 on CXCR1 and CXCR2 [33]. Thus, vCXCL1$_{GT1}$ recruits β-arrestins through CXCR1 and CXCR2, and vCXCL1$_{GT5}$ through

CXCR2. Together with the G protein pathway experiments, this confirms that vCXCL1$_{GT5}$ is a selective CXCR2 agonist.

## vCXCL1$_{GT5}$ does not induce migration through CXCR1 but attracts PBNs to the same degree as vCXCL$_{GT1}$

Another role of β-arrestin is to induce reorganization of the cytoskeleton thereby facilitating chemotaxis [34]. Having established that vCXCL1$_{GT5}$ activates and induces β-arrestin recruitment through CXCR2 but not CXCR1, while vCXCL1$_{GT1}$ was an agonist on the same pathways on both receptors, we investigated whether the same pattern was seen in chemotaxis. The chemotactic properties of vCXCL1$_{GT1}$ and vCXCL1$_{GT5}$ were first tested using the murine pre-B lymphocyte cell line L1.2 stably transfected with either CXCR1 or CXCR2 in a transwell migration assay (Fig 7A and 7B). The efficacy of vCXCL1$_{GT1}$ in inducing chemotaxis of CXCR1-expressing L1.2 cells was comparable to that of CXCL8, although CXCL8 was more potent on this receptor. In contrast, 1 μM vCXCL1$_{GT5}$ produced a minimal response at 0.2% of the efficacy of 1 nM CXCL8, which is not visible in Fig 7A. CXCL8 elicited the classical bell-shaped dose response, whereas concentrations higher than 1 μM of vCXCL1$_{GT1}$ would be necessary to evoke the downward slope of the curve. For the CXCR2-expressing cells, CXCL8, vCXCL1$_{GT1}$ and vCXCL1$_{GT5}$ all elicited the classical bell-shaped dose-response curves typical for chemotaxis assays. CXCL8 was again the most potent chemokine in evoking chemotaxis, while the viral chemokines were approximately equipotent, with vCXCL1$_{GT1}$ displaying the highest efficacy. Next, we compared the two viral chemokines in a migration assay using primary cells. CXCR1 and CXCR2 are primarily present on neutrophils and responsible for their migration to sites of inflammation in response to local chemokine gradients. Thus, a transwell cell migration assay was performed for CXCL8, vCXCL1$_{GT1}$ and vCXCL1$_{GT5}$ using isolated PBNs, which showed substantial migration for all three chemokines (Fig 7C). Importantly, the similar migratory response of neutrophils to vCXCL1$_{GT1}$ and vCXCL1$_{GT5}$ indicates that CXCR1 is not necessary for inducing migration of neutrophils. The higher efficacy of vCXCL1$_{GT1}$ compared to vCXCL1$_{GT5}$ in inducing migration of CXCR2 transfected cells—in contrast to the same efficacy for the two viral chemokines to induce migration of neutrophils —is likely due to cell dependent differences in the chemotactic machinery resulting in saturation of specific steps in the signaling cascade leading to migration.

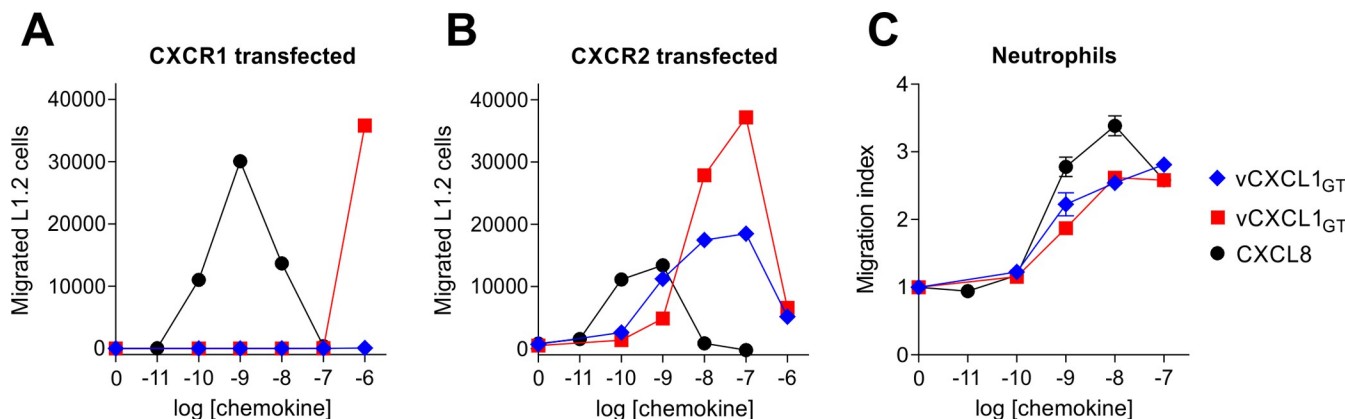

**Fig 7. vCXCL1$_{GT1}$, vCXCL$_{GT5}$ and CXCL8-induced migration of L1.2 cells stably expressing either CXCR1 or CXCR2, and neutrophils.** Chemotactic activities of L1.2 cells stably transfected with (**A**) CXCR1 (n = 4) and (**B**) CXCR2 (n = 3) toward vCXCL1$_{GT1}$, wild-type vCXCL$_{GT5}$ and CXCL8 in a transwell migration assay. Representative assays with error bars presented as SEM. (**C**) Chemotactic activity of human neutrophils toward vCXCL1$_{GT1}$, wild-type vCXCL1$_{GT5}$ and CXCL8 in a transwell migration assay, shown as migration index (MI) determined from point measurement divided by the background measurement (n = 4). Representative assay with error bars presented as SEM.

### Deletion of the C-terminus of vCXCL1$_{GT1}$ results in solvent exposure of chemokine core tryptophan residues and in a non-functional protein

In general, deletion of the C-terminus from human chemokines with long C-termini does not lead to loss of functional activity [21,22,35–44] (S1 Table). To investigate if this was also the case for vCXCL1$_{GT1}$, we expressed and purified a tailless vCXCL1$_{GT1}$ protein from *E. coli* where the 24 distal residues were deleted, resulting in a 71-residue protein (Fig 8A). Interestingly, the tailless vCXCL1$_{GT1}$ did not induce IP3 accumulation through CXCR1 and CXCR2 in contrast to the wild-type vCXCL1$_{GT1}$ (Fig 8B and 8C). Next, the thermal denaturation of purified wild-type vCXCL1$_{GT1}$ and the tailless vCXCL1$_{GT1}$ was assessed by nano differential

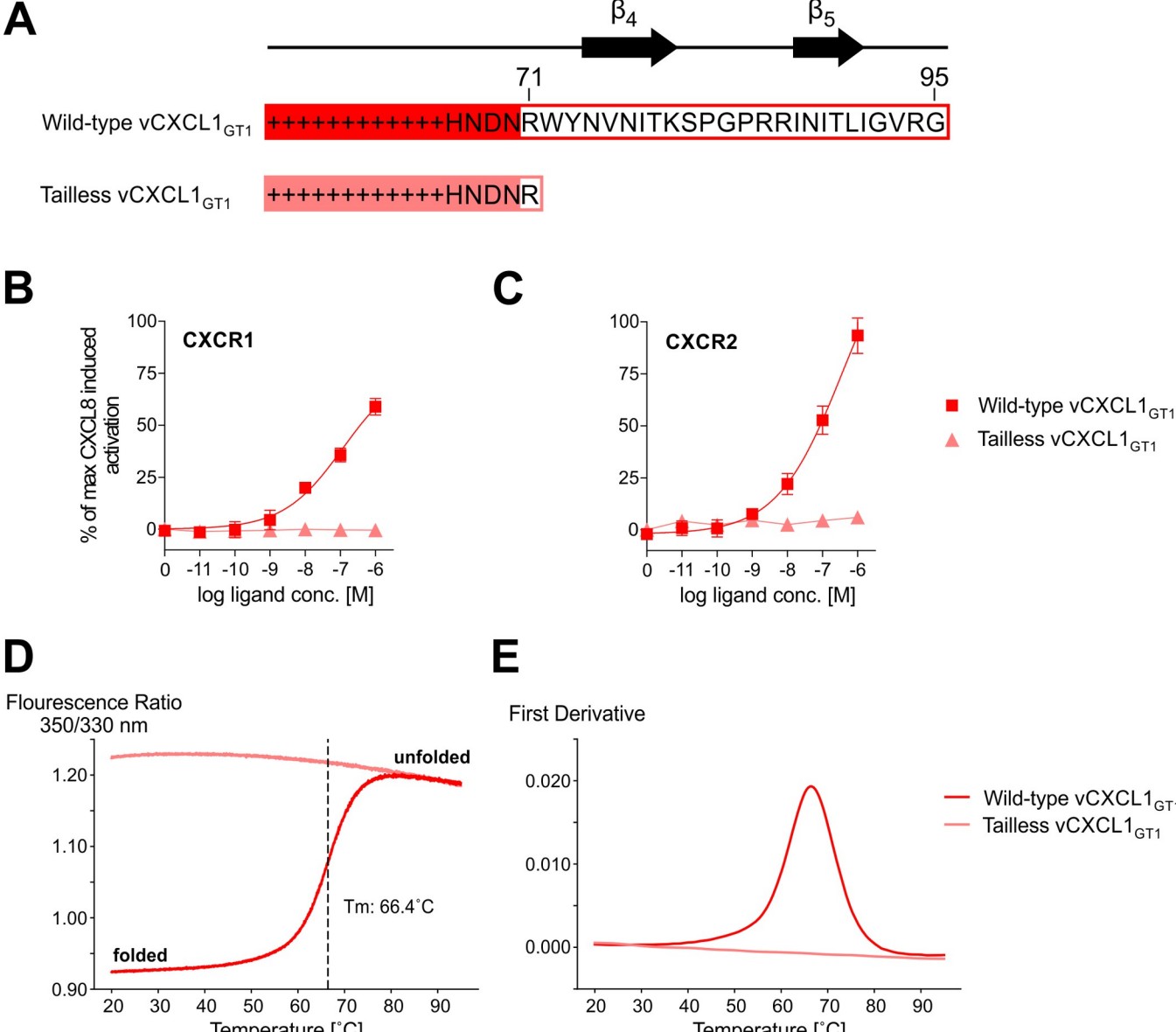

**Fig 8. Functional activity and thermal melting curves for tailless and wild-type vCXCL1$_{GT1}$.** (**A**) C-terminal sequences and secondary structure of wild-type and tailless vCXCL1$_{GT1}$. (**B+C**) G protein signaling in an IP accumulation assay of wild-type and tailless vCXCL1$_{GT1}$ on CXCR1 and CXCR2; n = 3. Error bars shown as SEM. (**D+E**) Thermal melting curves and their first derivative for wild-type and tailless vCXCL1$_{GT1}$; n = 3.

scanning fluorimetry (nanoDSF) (Fig 8D and 8E). Due to the high sensitivity of tryptophan sidechains to their chemical environment, protein unfolding can be detected by monitoring the intensity and ratio of emission at 350 nm to 330 nm [45]. This analysis gave a two-state melting curve with an apparent $T_m$ of 66.4˚C for wild-type vCXCL1$_{GT1}$, whereas tailless vCXCL1$_{GT1}$ was unfolded from 20–95˚C, demonstrating that the C-terminal β-hairpin of vCXCL1$_{GT1}$ is required for folding of a stable chemokine core.

## Discussion

The 14 stable UL146 genotypes encoded by circulating HCMV strains display considerable variation both among themselves and compared to human chemokines on several parameters such as length of the N-terminus, the presence or lack of an ELR motif, glycosylation pattern, and length of the C-terminus. We here show that the two non-ELR HCMV-encoded chemokines vCXCL1$_{GT5}$ and vCXCL1$_{GT6}$ are selective agonists for CXCR2—a property not shared by any human chemokines. Furthermore, Rosetta analysis predicted that the C-terminal tail of most vCXCL1 genotypes adopts a novel β-hairpin that is required for receptor activation, likely by stabilization of the chemokine fold.

### Structural elements of the vCXCL1$_{GT1}$ and vCXCL1$_{GT5}$ proteins

The predicted signal peptide cleavage sites of vCXCL1$_{GT1}$ and vCXCL1$_{GT5}$ were confirmed through eukaryotic chemokine expression using MALDI-TOF and PNGase F treatment (Fig 2), revealing different lengths of the N-terminus for ELR and non-ELR vCXCL1. In the initial report on vCXCL1, the molecular weight of the deglycosylated, secreted protein was estimated using western blots [6]. Another group expressed several vCXCL1 variants in insect cells, and although MALDI-TOF was used for mass determinations, molecular weights were not reported for the individual vCXCL1 genotypes, nor was removal of N- and O-linked glycans [10].

In the present study, we found vCXCL1$_{GT1}$ and vCXCL1$_{GT5}$ to be N-glycosylated as predicted (Fig 1). The pattern of predicted glycosylation for all genotypes is interesting and a distinctive feature for the individual genotypes. Six genotypes (GT1, 2, 3, 5, 8 and 12) are predicted to be N-glycosylated, five genotypes (GT4, 9, 10, 11 and 13) to be O-glycosylated and three genotypes (GT6, 7 and 14) not to be glycosylated at all (Figs 1 and 2A). The resulting proteoforms of these post-translational modifications provide the vCXCL1 variants with different epitopes important for binding to the extracellular matrix and immune cells and variation in the ability to shield epitopes of the chemokine core from the immune system of the host.

Homology modeling predicted that 11 of the 14 vCXCL1 genotypes form a novel 4th and 5th β-strand at the extended C-terminus. These strands form a β-hairpin that turns upon itself at a common glycine-proline motif (S1 Fig). The presence of this β-hairpin in multiple top scoring modeling runs in Rosetta and the identification of two stable β-strands by secondary NMR chemical shift analysis of five atoms ($H^N$, Cα, Cβ, CO, N) for the last 45 residues of vCXCL1$_{GT1}$ in TALOS analysis support the existence of this unique secondary structure. While an extended C-terminal tail is also found in some human chemokines, these lack secondary structure elements in contrast to these HCMV-encoded chemokines [19–22] (S1 Table). The presence of the β-hairpin in a majority of the 14 genotypes implies conserved but unknown functions. The thermal unfolding and functional data of the tailless vCXCL1$_{GT1}$ protein suggest that one role of the terminal β-hairpin is to stabilize the chemokine fold (Fig 8D and 8E). This is in contrast to NMR structures of tailless versions of human chemokines with extended C-termini (XCL1, CCL21, CCL25, CCL28 and CXCL12γ), which contain a preserved chemokine fold [21,22,35,39,40,46] (S1 Table) and which retain functional activity [21,22,35,36,39–44] (S1 Table). Furthermore, in addition to stabilization of the chemokine

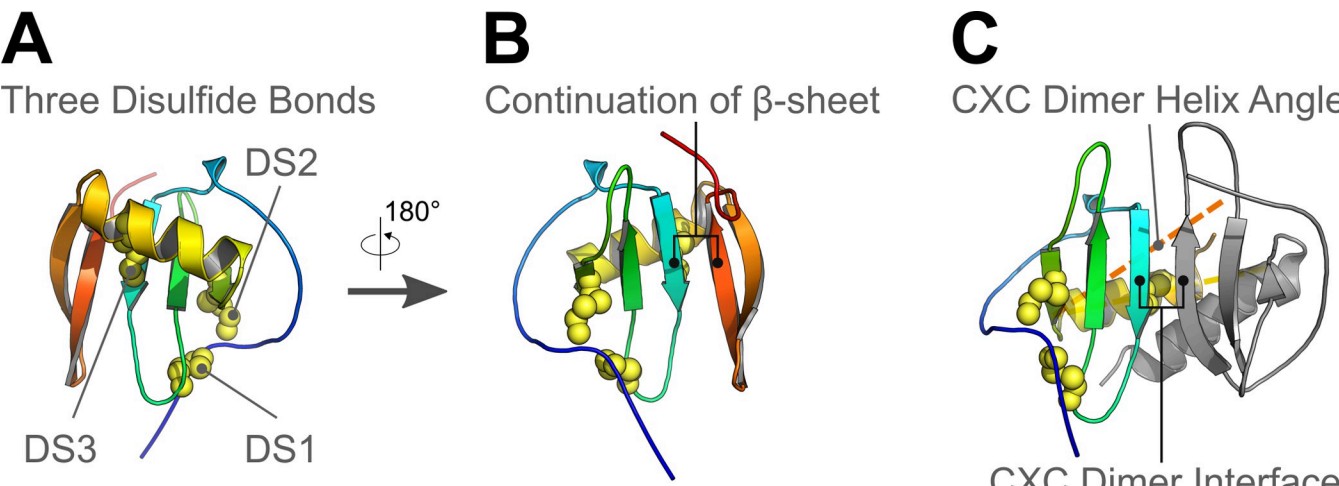

**Fig 9. The predicted third disulfide bond of vCXCL1$_{GT4}$ allows the C-terminal β-hairpin to mimic CXC chemokine dimerization. (A)** The top model of vCXCL1$_{GT4}$ is shown in cartoon form with disulfide bridges shown as spheres. The canonical disulfide bonds (DS1 and DS2) stabilize chemokine tertiary structure. A unique disulfide (DS3) links the β1-strand to the C-terminal α-helix. **(B)** The top model of vCXCL1$_{GT4}$ is rotated 180˚ to show the front view. vCXCL1$_{GT4}$ models have a consensus location for the C-terminal β-hairpin, which continues the 3-stranded β-sheet core. This mimics the CXC dimer interface shown in panel C. The constraint from the third disulfide bond may help stabilize this conformation, as there was no consensus β-hairpin location in other genotype models. **(C)** The top vCXCL1$_{GT4}$ model was aligned to one of the CXCL8 chains in a structure of a CXCL8 dimer (PDB ID: 6LFM). The other dimer subunit is shown in cartoon form (gray), and the extended C-terminus of vCXCL1$_{GT4}$ is hidden for clarity as it overlaps with the CXC dimer interface. The third disulfide bond in vCXCL1$_{GT4}$ locks the α-helix into an orientation that would clash with a classic CXC dimer. The vCXCL1$_{GT4}$ α-helix angle varies 20˚ from the α-helix of CXCL8, shown as a dotted orange line.

fold, other functions of the β-hairpin are likely. Documented roles for the unstructured extended C-termini of human chemokines include antifungal activity [21] and glycosamino-glycan (GAG) binding [22,40,43]. Thus, a high content of basic residues in the C-terminus has been linked to GAG binding. The C-termini of CCL21 and CXCL12γ are highly basic (32% and 58%) and deletion of these C-termini results in loss of GAG binding [22,40,43] (S1 Table). Likewise, the C-termini of vCXCL1$_{GT10}$ (34%) and vCXCL1$_{GT12}$ (27%) also have a high content of basic residues suggesting GAG binding (S1 Table). In contrast, the extended C-termini of some of the human (XCL1 and CCL16) and some of the viral (vCXCL1$_{GT1}$, vCXCL1$_{GT2}$ and vCXCL1$_{GT4}$) chemokines are significantly less basic, which suggest binding to unidentified proteins such as cell surface receptors or other chemokines (S1 Table). Chemokine binding is a particularly interesting possibility as CXC chemokines are known to dimerize along the β1-strands. This possibility was illustrated in models of vCXCL1$_{GT4}$ that have a non-canonical third disulfide bond which was not found in other genotypes (Figs 1 and 9A). This disulfide bond linked the first β-strand to the end of the α-helix and facilitated the β-hairpin to continue the anti-parallel β-sheet core (Fig 9B). This organization mimics CXC dimerization, as seen by alignment to one monomer of a CXCL8 dimer (Fig 9C). While it is possible that other geno-types may adopt this extended β-sheet conformation, it is less likely as the α-helix is unre-strained by a disulfide bridge, thus allowing the β-hairpin more freedom of movement. Importantly, this conformation highlights the possibility for the β-hairpin to bind the β1-strand of another chemokine. Binding host chemokines and inhibiting their interaction with cognate receptors could manipulate the immune response and lead to improved viral fitness.

## The non-ELR vCXCL1$_{GT5}$ chemokine is a selective CXCR2 agonist

We have previously established that the ELR chemokine vCXCL1$_{GT1}$ (encoded by UL146 from the Toledo strain) is a dual CXCR1 and CXCR2 agonist when tested on a panel of cell lines

expressing all human chemokine receptors individually [9]. In the present study, we elaborate on that finding by demonstrating that the non-ELR vCXCL1$_{GT5}$ (encoded by the Davis strain) is a selective CXCR2 agonist in G protein signaling using an IP accumulation assay (Figs 4B, 5A, and 5B). Thus, both vCXCL1 variants have the common trait of only targeting receptors activated by human ELR chemokines. The importance of CXCR1 and CXCR2 as receptor targets for this HCMV-encoded family of chemokines is further supported by the fact that both vCXCL1$_{GT4}$ and vCXCL1$_{GT6}$ activated CXCR2 selectively (Fig 5B and 5D). As some viral chemokines, e.g. vMIP-II encoded by Kaposi's sarcoma-associated herpesvirus and MC148 encoded by a poxvirus, act as chemokine receptor antagonists [47–49], vCXCL1$_{GT5}$ was screened for antagonism on the 18 chemokine receptors but was not found to block G protein signaling (Fig 4C). The CXCR2 selectivity of vCXCL1$_{GT5}$ was confirmed for the β-arrestin pathway in a BRET-based assay, where it was unable to stimulate β-arrestin recruitment through CXCR1 (Fig 6C), while potent recruitment was seen through CXCR2 (Fig 6F). In contrast, vCXCL1$_{GT1}$ recruited β-arrestin through both CXCR1 and CXCR2, thus confirming its action as a dual agonist. Finally, the selective use of CXCR2 for vCXCL1$_{GT5}$ was also confirmed in migration assays using stable cell-lines expressing either CXCR1 or CXCR2. Although the non-ELR vCXCL1$_{GT5}$ only activated CXCR2, we found that it was able to induce chemotaxis of neutrophils with equal potency and efficacy to the ELR vCXCL1$_{GT1}$ in a transwell migration assay (Fig 7). Another group previously expressed all vCXCL1 genotypes except genotypes 3, 4, and 5 in insect cells and tested them on PBNs and cell lines transfected with CXCR2 (but not CXCR1) in different functional assays [10]. Although vCXCL1$_{GT5}$ was not included, the other non-ELR genotype vCXCL1$_{GT6}$ with a highly similar sequence to genotype 5 (Fig 1) was tested. They found that vCXCL1$_{GT1}$ was able to activate G protein-dependent and–independent pathways through CXCR2, agreeing with our observation, but in contrast to our results, they found vCXCL1$_{GT6}$ unable to stimulate G protein signaling and β-arrestin recruitment through CXCR2 [10]. Additionally, although they and others have reported that the ELR-containing vCXCL1$_{GT1}$ and vCXCL1$_{GT2}$ elicit robust chemotactic responses in neutrophils [6,10,11], no induction of neutrophil chemotaxis was observed for vCXCL1$_{GT6}$ [10]. Thus, in contrast to our findings for vCXCL1$_{GT6}$ and the analogous non-ELR chemokine vCXCL1$_{GT5}$, they did not see any activity of vCXCL1$_{GT6}$. There could be various reasons for this discrepancy, one being expression of the non-ELR vCXCL1 with a short N-terminus, corresponding to our N-terminally truncated vCXCL1$_{GT5 (6–97)}$, which we here show is not the mature, functional chemokine.

Herpesviruses not only interact with the human chemokine system through virally encoded chemokines, but also through virally encoded chemokine receptors. For example, HCMV encodes four 7TM (seven-transmembrane domain) receptors including US28, which is a well-characterized chemokine receptor [2,3]. Cytomegalovirus-infected cells express US28 during latency and the receptor has been validated as a drug target in mouse models [50]. US28 can signal constitutively but also through binding of CC and CX$_3$C chemokines [51,52]. While no current data supports US28 interaction with CXC chemokines, we speculated whether HCMV could activate US28 through vCXCL1$_{GT5}$ in an autocrine loop, and thereby rewire an infected cell to a state supporting latency. However, we were unable to demonstrate US28 activation by vCXCL1$_{GT5}$ (Fig 4B).

The recently solved structure of CXCL8 bound to CXCR2 (PDB ID: 6LFO) has shed light on receptor activation by an ELR chemokine [53]. This structure shows extensive contacts between the receptor and the ELR motif considered crucial for activation. However, here we show that vCXCL1$_{GT5}$ and vCXCL1$_{GT6}$, both of which lack an ELR motif, signal through CXCR2, and to our knowledge, this has not been described for any naturally occurring chemokine before. To investigate the ability of vCXCL1$_{GT5}$ to circumvent the ELR selectivity of

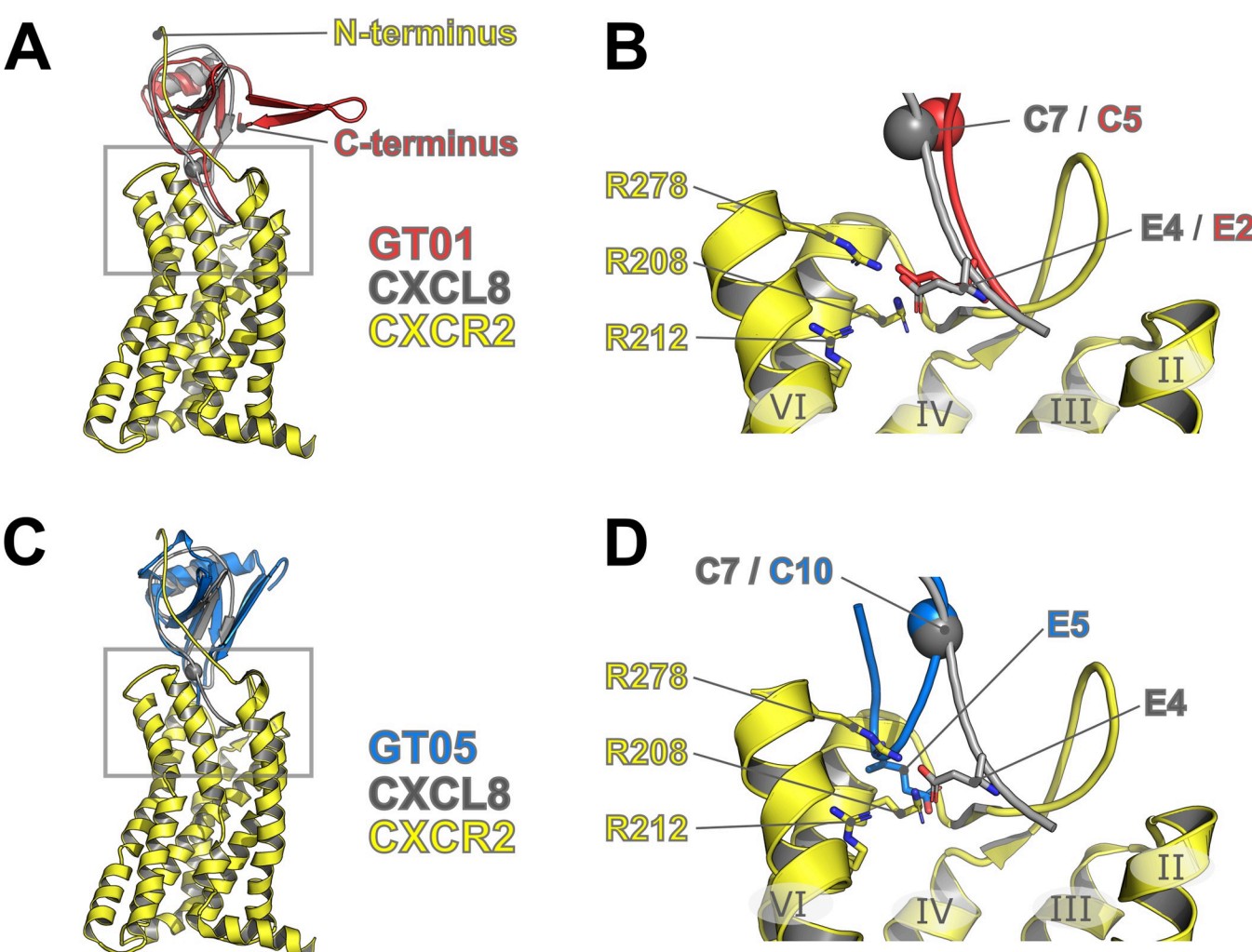

**Fig 10. Alignment of vCXCL1$_{GT1}$ and vCXCL1$_{GT5}$ models with CXCL8 in the experimental CXCL8:CXCR2 complex. (A)** The top model (lowest Rosetta total score) for vCXCL1$_{GT1}$ is shown in cartoon form (red) aligned to the structure of the CXCL8:CXCR2 complex (PDB: 6LFO). The extended C-terminus is unlikely to interfere with binding to CXCR2, as the N-terminus of CXCR2 (chemokine-recognition site 1) binds the opposite side of the chemokine. **(B)** Expanded view of the boxed region in panel A showing important residues as sticks and numbered by residues present in the structure. The first conserved cysteine residue is shown as a sphere. The glutamate in the ELR motif is in a very similar position for both vCXCL1$_{GT1}$ and CXCL8. This residue has been shown to be critical for the activation of CXCR2 and interacts with receptor residues R208, R212, and R278. **(C)** The top scoring model for vCXCL1$_{GT5}$ is shown in cartoon form (blue) aligned to the structure of the CXCL8:CXCR2 complex. The β-hairpin is not predicted to interrupt site-1 binding in this model. **(D)** Expanded view of the boxed region in panel C showing important residues as sticks and numbered by residues present in the structure. The first conserved cysteine residue is shown as a sphere. vCXCL1$_{GT5}$ does not contain an ELR motif, instead the five residues proceeding the cysteine are 5-EGNGR-9 (Glu5 shown in stick form). The longer N-terminus in vCXCL1$_{GT5}$ is an unstructured loop in this model, which was generated without the receptor present. In this position, it is unlikely that Glu5 will make sufficient contact with the displayed receptor residues. During a binding event, the chemokine N-terminus may adopt a stable conformation, facilitated by Arg9, whereby Glu5 could participate in receptor activation. Alternatively, Asn7 may form ELR-like contacts sufficient to activate the receptor.

CXCR2, we aligned our chemokine models to CXCL8 in the CXCL8:CXCR2 complex (Fig 10). Both vCXCL1$_{GT1}$ and vCXCL1$_{GT5}$ mimic the core structure of CXCL8, and the C-terminal β-hairpin does not appear to hinder interactions with the receptor N-terminus (chemokine-recognition site 1) (Fig 10A and 10C). The N-terminus of the chemokine, which contains the ELR motif, binds and activates the receptor via the orthosteric pocket. The model of vCXCL1$_{GT1}$ has the N-terminus in a matching position as CXCL8, with Glu2 (Glu4 in this CXCL8 structure) of the ELR motif located near receptor residues Arg208, Arg212, and Arg278 that drive

receptor activation (Fig 10B). In lieu of the ELR motif, which precedes the first conserved cysteine, vCXCL1$_{GT5}$ contains the N-terminal sequence 1-GFGTEGNGR-9. In the top models of vCXCL1$_{GT5}$ (generated without the receptor present), this longer N-terminus turns back upon itself and is not likely to activate CXCR2 in the same manner as CXCL8 (Fig 10D). During a binding event, the N-terminus of vCXCL1$_{GT5}$ may adopt a stable conformation, facilitated by Arg9 at the Cys-1 position which acts as a stabilizing anchor point in CXCL8. It is unclear if Asn7 is sufficient to activate CXCR2, or if Glu5 can be reoriented to contact receptor residues critical for activation. The truncation data for vCXCL1$_{GT5}$ support the first scenario, as removal of 1-GFGTE-5 reduces, but does not completely abrogate, signaling ability. Similarly, removal of 1-GFG-3 reduces signaling ability. These residues may act to stabilize the extended N-terminus in an orientation favorable for activation.

Demonstrating that some circulating HCMV strains encode a selective CXCR2 agonist instead of a CXCR1/CXCR2 dual agonist presents the question of how the virus can benefit from this receptor selectivity. Neutrophils are known to have high expression of both CXCR1 and CXCR2 [13] which HCMV exploits through vCXCL1 to attract, infect, and disseminate [6,9,12]. Others have proposed that the primary receptor for this action is CXCR2, as neutrophils migrate more effectively than CXCR1$^{+}$ CXCR2$^{-}$ NK cells in response to vCXCL1$_{GT2}$ [11]. Our data supports the latter theory, as no difference in the migration of neutrophils was observed between vCXCL1$_{GT1}$ (targeting CXCR1 and CXCR2) and the CXCR2 selective vCXCL1$_{GT5}$ (Fig 7). Additionally, this suggests that non-ELR vCXCL1 is a specific chemoattractant for neutrophils as its lack of CXCR1 activity would eliminate the unwanted attraction of CXCR1$^{+}$ CXCR2$^{-}$ leukocytes such as CD8$^{+}$ cytotoxic T-cells and NK cells and the subsequent killing of infected cells [13,54,55].

## Why do circulating HCMV strains encode 14 different forms of the same protein?

A possible explanation to this conundrum is that each UL146 genotype has developed in smaller isolated human communities and that these HCMV strains later fed into larger populations because of migration and trade. However, if this was the case it is difficult to explain that the same phenomenon is not seen for other human viruses encoding chemokines, including Kaposi's sarcoma-associated herpesvirus (KSHV), human herpesvirus 6 and 8 (HHV-6 and HHV-8), and the molluscum contagiosum virus [47–49,56–59]. Another possible explanation is that encoding several forms of the same immune modulator provides HCMV with the tools to successfully infect and establish infection in a broader variety of hosts, overcoming immunological bottlenecks (Fig 11). In that way the virus could ensure a large reservoir among humans with a diverse genetic and immunological makeup. This hypothesis can be investigated in different ways. Firstly, as in the present study by comparing variant vCXCL1 chemokines in biochemical/immunological assays, or by using *in vivo* models [12]. Secondly, by searching for associations between specific viral genotypes and CMV disease, which has the potential to show the effect of the gene variants if the lack of large sample sizes and controls groups are addressed. Lastly, by performing genome-wide association studies (GWAS) of human hosts infected with HCMV to identify host and viral gene variants associated with disease development.

In conclusion, we here identify two non-ELR chemokines encoded by HCMV UL146 genotype 5 and 6 capable of activating CXCR2. Furthermore, we present evidence that the majority of the 14 genotypic vCXCL1 chemokines–in contrast to human chemokines—adopt a unique C-terminal β-hairpin, with the function of stabilizing the chemokine fold. Future structural and cell biological studies of the UL146 gene products are required to determine possible

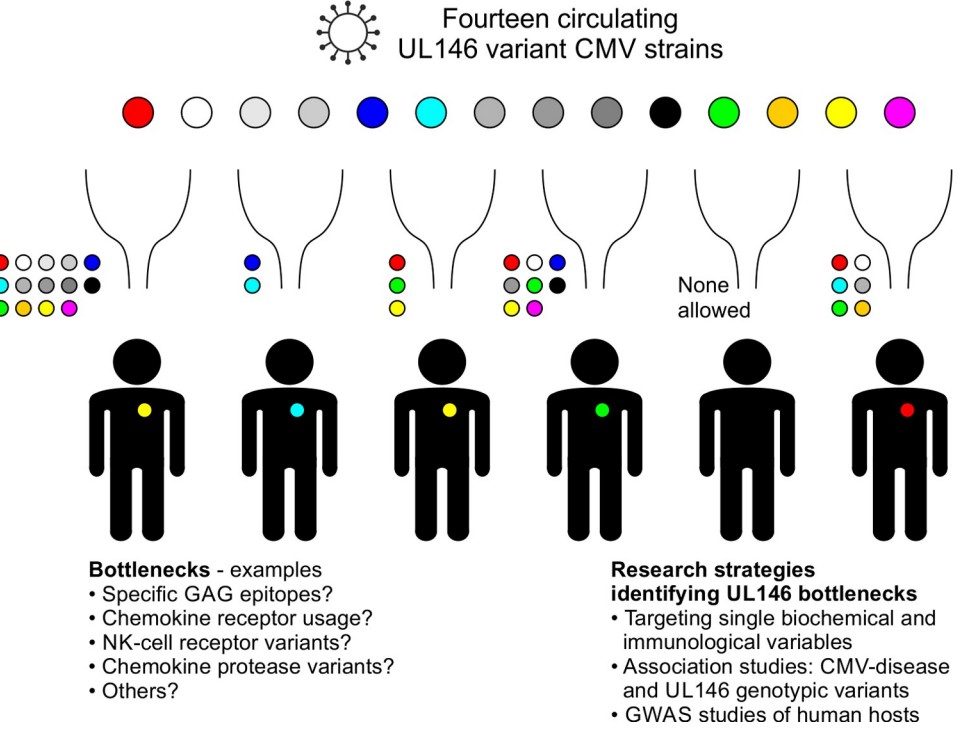

**Fig 11. Hypothetical model explaining the existence of 14 stable UL146 genotypes by bottlenecks in the genetic makeup of the host.** Unidentified bottlenecks in the host allowing only one or some of several possible UL146 genotypes among circulating HCMV strains to establish infection and disseminate in a given host. Small circles show which UL146 strains are able to pass the bottleneck. Note that *i)* different bottlenecks can allow infection from between one and 14 genotypes, and *ii)* a given host/HCMV strain pair can be the result of different bottlenecks (yellow virus on human that could potentially be infected by multiple different strains). Bottom left: examples of human proteins representing potential bottlenecks. Bottom right: examples of possible research strategies for identifying host bottlenecks.

interaction partners of the C-terminal β-hairpin and unravel other functions of this unique group of highly diverse chemokines.

## Materials and methods

### Plasmids used in cell-biological assays

For all chemokine receptors used, the human wild type receptor cDNAs without tags were cloned into the expression vector pcDNA3.1(+) (Invitrogen). The following restriction sites were used for insertion: CXCR1, CXCR2, CXCR3, CXCR4, CXCR6, XCR1, CX3CR1, CCR2, CCR6 and CCR7 (HindIII-XhoI); CXCR5 and CCR5 (HindIII-BamHI); CCR1, CCR3, CCR4, CCR9 and CCR10 (EcoRI-XhoI); CCR8 (HindIII-EcoRI); US28 (EcoRI-NotI). All constructs were verified by restriction endonuclease digestion and DNA sequencing. The chimeric G protein GαΔ6qi4myr (Gqi4myr) inserted into pcDNA3.1(+) was kindly provided by Evi Kostenis (University of Bonn) [60]. The Rluc8-arrestin3-Sp1 and mem-citrine-SH3, GRK2, and GRK6 plasmids were kindly provided by Jonathan Javitch (Columbia University) [61]. Receptor expression was verified by a functional response, i.e. when transfected cells responded upon stimulation by a natural chemokine agonist of the expressed chemokine receptor (for details see Fig 4) while non-transfected cells showed no response to the same chemokine.

## Chemokines

The human chemokines CCL1, CCL5, CCL7, CCL11, CCL17, CCL19, CCL20, CCL25, CCL27, CXCL8, CXCL11, CXCL12, CXCL13, CXCL16, and $CX_3CL1$ were purchased from PeproTech, whereas XCL1 was purchased from R&D Systems. All chemokines were reconstituted in a buffer containing 0.1% (w/v) bovine serum albumin (BSA) and 1 mM acetic acid.

## Cell lines and transfections

COS-7 and HEK293 cells (ATCC) were grown at 10% $CO_2$ and 37˚C in DMEM1885 supplemented with 10% (v/v) fetal bovine serum (FBS), 2 mM L-glutamine, 180 units/ml penicillin, and 45 μg/ml streptomycin. Transient transfection of COS-7 and HEK293 cells was performed by a chloroquine based calcium phosphate precipitation method, where DNA was diluted in TE buffer and $CaCl_2$ was added to a concentration of 0.25 M. This mixture was then slowly mixed with an equal volume of 2x HEPES buffered saline (HBS) and incubated for 45 min at RT. After incubation, the DNA mixture was dripped directly onto the cells, and growth medium containing 60 μg/ml chloroquine was added. The transfection was stopped after 5 h by changing medium.

CXCR1 (HindIII-EcoRI) and CXCR2 (HindIII-EcoRI) were inserted into the pTej vector [62] and transfected into the murine pre-B cell lines L1.2 by electroporation. Stable transfectants were obtained after limiting dilution and chemical selection with G418.

CHO-K1 (ATCC) and stable L1.2 cells were grown at 5% $CO_2$ and 37˚C in RPMI 1640 supplemented with 10% (v/v) FBS, 2 mM GlutaMAX (Gibco), 180 units/ml penicillin and 45 μg/ml streptomycin. Transient transfection of CHO-K1 cells was performed by lipofection using Lipofectamine 2000 (Invitrogen) according to manufacturer instructions.

## Eukaryotic expression and purification of UL146

For eukaryotic expression, the UL146 gene sequences from the Toledo strain (GenBank accession no. GU937742.2) and from the Davis strain (GenBank accession no. JX512198.1) were codon optimized by the JCat algorithm [27]. The genes were synthesized by GeneArt (Invitrogen) and inserted into the pcDNA3.1(+) expression vector (Invitrogen) using the BamHI and EcoRI restriction sites.

A method previously described for MC148 and vCCL3 was applied with minor modifications [48,59]. COS-7 cells were seeded in T175 flasks at 2 x $10^6$ cells per flask and transfected with UL146 or empty vector the following day. The day after transfection, growth medium was removed and cells were washed with PBS. 20 ml harvest medium was added (DMEM1880 with $NaHCO_3$, 2 mM L-glutamine, 180 units/ml penicillin, and 45 μg/ml streptomycin) and cells were incubated at culturing conditions for 24 h. The medium was collected, centrifuged at 500 $g$ for 20 min, and the supernatant was transferred to a clean container. Then the supernatant was adjusted to pH 4.5 and filtered through 0.22 μm PES filters (Corning) and the filtrate was diluted with Milli-Q water 1:1 to reduce ionic strength. Cation exchange columns were prepared by first adding 750 μl thoroughly mixed SP Sepharose FF resin two times, and each column was in turn washed with 5 ml Milli-Q water, 5 ml 1 M NaOH, and five times with 1 ml Milli-Q water or until achieving neutral pH of the pass-through. The columns were then washed with 5 ml 30% isopropanol and then five times with 1 ml Milli-Q water, before they were equilibrated two times with 5 ml 50 mM sodium acetate at pH 4.5. Afterwards, the diluted cell medium filtrate was loaded on the prepared cation exchange columns and allowed to pass before washing twice with 10 ml 50 mM sodium acetate. The bound protein was eluted by adding 1 ml 50 mM sodium acetate with 2 M NaCl at a time five times. This was repeated on the next day after allowing the cell cultures to recover for 5h with growth medium and then

changing back to harvest medium. The eluate was adjusted to pH 4 in 0.1% triflouroacetic acid (TFA), filtered, and loaded on a Resource 15RPC 3 ml column (GE Healthcare) for reverse-phase HPLC on an ÄKTApurifier (GE Healthcare) and the protein was eluted with a gradient of $CH_3CN$ from 0.1% TFA in water.

Purified chemokines were confirmed to elicit correct masses by matrix-assisted laser desorption-ionization time of flight (MALDI-TOF) analysis using an Autoflex II instrument (Bruker). All spectra were analyzed in the Bruker Daltonics flexAnalysis 2.2 software. Chemokines produced in eukaryotic cells were further confirmed by peptide mass fingerprint using trypsin digestion (Roche Applied Science) according to the manufacturer's instructions. Removal of N-glycans from chemokines produced in eukaryotic cells was done using PNGase F treatment according to manufacturer's instructions (Sigma-Aldrich).

The following gene sequences were used for eukaryotic protein expression:

*Genotype 1 UL146 (Toledo strain)*

ATGCGGCTGATCTTCGGCGCCCTGATCATCTTCCTGGCCTACGTGTACCACTAC
GAAGTGAACGGCACCGAGCTGAGATGCCGGTGCCTGCACAGAAAGTGGCCCCCCA
ACAAGATCATCCTGGGCAACTACTGGCTGCACCGGGACCCTAGAGGCCCTGGCTG
CGACAAGAACGAGCATCTGCTGTACCCCGACGGCCGGAAGCCTCCTGGACCTGGC
GTGTGTCTGAGCCCCGATCACCTGTTCAGCAAGTGGCTGGACAAGCACAACGACA
ACCGGTGGTACAACGTGAACATCACCAAGAGCCCTGGCCCCAGACGGATCAACAT
CACCCTGATCGGCGTGCGGGGCTGA

*Genotype 5 UL146 (Davis strain)*

ATGCGGCTGATCTTCGGCCTGCTGATCATCTTCATCGTGACCGATACCTGCAAC
GGCGGCTTCGGCACAGAGGGCAACGGCAGATGTACCTGCATCGGCTACCACCGG
CTGCTGGGCCAGCTGCCTAGAGGCACATTTTGGCTGGGACATCTGCCCCCTGGCA
GCCACTGTCCTAAGGGCCAAGTGATGATCAAGATCGGCCAGGGCCCCATCGTGTG
CCTGAGCGATTACCACCCCCTGAGCAAGTGGATGTACGGCAACCACAAGAGCGG
CAGCGAGACATGGCTGCAGATCAAGATGGAAGGCCCCAGAAACGCCACCGTGGT
GCAGCGGAGCAACACCAGACCTTGA

## Prokaryotic expression and purification of UL146 –Protocol 1 (Copenhagen)

For prokaryotic expression, the sequences for the mature genotype 1, genotype 5 (1–97), genotype 5 (4–97), and genotype 5 (6–97) UL146 chemokines were codon-optimized for *E. coli* expression and synthesized by Integrated DNA Technologies, and inserted into the prokaryotic expression vector pET-21a(+) (provided by Mads Gravers Jeppesen from Synklino; www. Synklino.com)) using the restriction sites NdeI and HindIII.

BL21 Star (DE3) pLysS OneShot competent cells (Invitrogen) were transformed with UL146 in the pET-21a(+) and cultured according to the manufacturer's instructions at 37˚C in lysogeny broth (LB) medium with added 100 μg/mL ampicillin and 25 μg/mL chloramphenicol. A single colony was picked and inoculated in 5–200 mL of LB medium and grown overnight at 37˚C. The next morning, the culture was diluted 1:100 in LB medium and grown at 37˚C until an optical density of 0.8 at 600 nm wavelength was achieved, measured on a NanoDrop 2000c Spectrophotometer (Thermo Fisher). Protein expression was induced by adding 0.5 mM isopropyl-D-1-thiogalactopyranoside (IPTG), and after 3 hours of post induction growth, the cells were collected by centrifugation for 30 min at 5000 *g*. The supernatant was removed and the cell pellet was either used immediately or flash-frozen with liquid nitrogen and stored at -80˚C.

The cells were ruptured by sonication with 20 s on and off pulses at 40% amplitude for 3 min using the Misonix Ultrasonic Liquid Processor The Sonicator 4000 with a 4.8 mm tip

diameter in the following buffer: 50 mM trisaminomethane (Tris)-HCl pH 8, 100 mM NaCl, 5 mM EDTA, 10 mM MgCl$_2$, 2 mM dithiothreitol (DTT), 1 mM phenylmethane sulfonyl fluoride. The suspension was centrifuged for 30 min at 15000 *g*, the supernatant was discarded and the pellet was resuspended and washed 4–8 times with 50 mM Tris-HCl pH 8, 300 mM NaCl, 0.1% Triton X-114, and 5 mM DTT. Then, 20 mL of denaturation buffer (3 M Guanidinium chloride (GdmCl), 100 mM Tris-Cl pH 8, 5 mM EDTA, and 5 mM DTT) per 1 g of pellet was added to solubilize the inclusion bodies, followed by 1 h incubation at RT and centrifugation for 30 min at 15000 *g*. The inclusion bodies were dialyzed at 4˚C against 0.5x PBS (pH 6.5) containing 0.2 mM cystine and 1 mM cysteine overnight and then again for 3 h. The protein sample was centrifuged for 30 min at 5000 *g*, filtered through 0.22 μm filters, and loaded onto a 1–15 mL HiTrap SP FF column equilibrated in buffer A (50 mM PBS pH 6.5) and cation exchange chromatography was performed using the ÄKTA Pure (GE Healthcare) FPLC instrument. The bound protein was eluted either by a linear gradient 0–100% over 20 column volume of buffer B (50 mM PBS pH 6.5, 1 M NaCl). The fractions with the protein of interest were concentrated with a Vivaspin 20 5000 MWCO PES ultrafiltration unit (Sartorius) by manufacturer's instructions, and afterwards loaded onto a size-exclusion chromatography (SEC) XK16/600 Superdex 75PG column, equilibrated in 1x PBS with adjusted pH to 6.5. The fractions with the protein of interest were concentrated by ultrafiltration, and concentrations were quantified by measuring the absorption at 280 nm wavelength with the NanoDrop 2000c Spectrophotometer. SDS-PAGE, coomassie staining, densitometry, and endoxtoxin level measurements were carried out by standard protocols for quality control. The samples were aliquoted, flash-frozen in liquid nitrogen, and stored at -80˚C for further use.

Purified chemokines were confirmed to elicit correct masses by matrix-assisted laser desorption-ionization time of flight (MALDI-TOF) similarly to for the eukaryotic protein expression. The identities of the purified chemokines were furthermore confirmed by peptide mass fingerprint using LysC digestion (Roche Applied Science) according to the manufacturer's instructions.

The following gene sequences were used for expression in protocol 1:

*Genotype 1 UL146 (Toledo strain)*

ATGACGGAACTTCGTTGCCGTTGCCTTCACCGCAAATGGCCGCCAAACAAAATT
ATCTTGGGTAACTACTGGTTGCATCGCGACCCACGCGGACCTGGCTGCGATAAGA
ATGAACACTTATTATACCCGGACGGGCGCAAACCCCCAGGCCCTGGGGTGTGCTT
ATCACCAGATCATTTGTTTTCCAAGTGGCTGGACAAACACAATGACAACCGCTGGT
ATAACGTCAACATCACGAAATCTCCTGGTCCACGTCGTATTAACATCACACTTATC
GGGGTGCGCGGCTAA

*Genotype 5 UL146 (Davis strain, 1–97)*

ATGGGATTCGGAACTGAGGGTAACGGTCGCTGTACCTGTATTGGGTACCACCGC
CTGCTTGGTCAGCTTCCTCGCGGAACCTTCTGGTTAGGCCACCTTCCGCCAGGGTC
CCACTGTCCGAAGGGCCAGGTGATGATTAAGATCGGTCAAGGCCCCATCGTTTGCT
TGTCCGATTATCACCCTTTGAGTAAATGGATGTATGGGAACCACAAGAGCGGATCT
GAAACGTGGTTGCAGATTAAGATGGAAGGACCGCGCAACGCCACTGTGGTGCAAC
GTTCTAATACACGTCCGTAA

*Genotype 5 UL146 (4–97)*

ATGACTGAGGGTAACGGTCGCTGTACCTGTATTGGGTACCACCGCCTGCTTGGT
CAGCTTCCTCGCGGAACCTTCTGGTTAGGCCACCTTCCGCCAGGGTCCCACTGTCC
GAAGGGCCAGGTGATGATTAAGATCGGTCAAGGCCCCATCGTTTGCTTGTCCGAT
TATCACCCTTTGAGTAAATGGATGTATGGGAACCACAAGAGCGGATCTGAAACGT
GGTTGCAGATTAAGATGGAAGGACCGCGCAACGCCACTGTGGTGCAACGTTCTAA
TACACGTCCGTAA

*Genotype 5 UL146 (6–97)*

ATGGGTAACGGTCGCTGTACCTGTATTGGGTACCACCGCCTGCTTGGTCAGCTT
CCTCGCGGAACCTTCTGGTTAGGCCACCTTCCGCCAGGGTCCCACTGTCCGAAGGG
CCAGGTGATGATTAAGATCGGTCAAGGCCCCATCGTTTGCTTGTCCGATTATCACC
CTTTGAGTAAATGGATGTATGGGAACCACAAGAGCGGATCTGAAACGTGGTTGCA
GATTAAGATGGAAGGACCGCGCAACGCCACTGTGGTGCAACGTTCTAATACACGT
CCGTAA

## Prokaryotic expression and purification of UL146 –Protocol 2 (Wisconsin)

Sequences of vCXCL1 genotypes 1, 4, 5, and 6 were codon-optimized for *E. coli* expression and ordered from GenScript. Constructs were cloned into a pET28a-6xHis-SUMO3 vector and expressed in BL21 DE3 *E.coli*. Cells were expressed at 37˚C in LB medium and induced with 1 mM isopropyl-β-D-thiogalactopyranoside (IPTG) at an OD600 of 0.6. Cultures continued to grow for 5 and a half hours before bacteria were pelleted by centrifugation and stored at -20˚C. For uniform labeling with $^{15}$N, cells were grown in M9 media containing $^{15}$N-ammonium chloride as the sole nitrogen source. Bacterial pellets were resuspended in ~20 mL of Buffer A (50 mM $Na_2PO_4$ (pH 8.0), 300 mM NaCl, 10 mM imidazole, 1 mM phenylmethylsulphonyl fluoride (PMSF), and 0.1% (v/v) 2-mercaptoethanol (BME)) per pellet and lysed via sonication. Lysed cells were clarified at 15,000 x g and the supernatant was discarded. Pellets were resuspended by sonication in ~20 mL of Buffer AD (6 M guanidinium, 50 mM Na2PO4 (pH 8.0), 300 mM NaCl, 10 mM imidazole) and spun down at 18,000 x g for 20 min. Using an AKTA-Start system (GE Healthcare), supernatant was loaded onto a Ni-NTA column equilibrated in Buffer AD. The column was washed with Buffer AD, and proteins were eluted using Buffer BD (6 M guanidinium, 50 mM sodium acetate (pH 4.5), 300 mM NaCl, and 10 mM imidazole). Proteins were refolded overnight via drop-wise dilution into a 12-fold greater volume of Refold Buffer (50 mM Tris (pH 7.6), 50 mM NaCl) with the addition of 20 mM cysteine, and 0.5 mM cystine. Refolded protein was concentrated in an Amicon Stirred Cell concentrator (Millipore Sigma) using a 10 kDa membrane. Concentrated protein was added to 3.5 kDa dialysis tubing with the addition of ULP1 to cleave the N-terminal 6xHis-SUMO3-tag and dialyzed at 25˚C against Refold Buffer overnight. The AKTA-Start system was used to load cleaved proteins onto a SP FastFlow cation exchange column equilibrated in Refold Buffer. The column was washed with Refold Buffer, and proteins were eluted using Elution Buffer (50 mM Tris (pH 7.6), 1 M NaCl). vCXCL1 proteins were separated from reduced or aggregated species by reversed-phase high performance liquid chromatography, lyophilized, and stored at -80˚C for further use. Purity and identity of all proteins were confirmed by electrospray ionization mass spectrometry using a Thermo LTQ instrument, and SDS-PAGE with Coomassie staining.

Recombinant proteins of vCXCL1$_{GT1}$ and vCXCL1$_{GT5}$ were produced by both protocol 1 and 2, and when compared, they activated G proteins through CXCR1 and CXCR2 to exactly the same extent in the IP3 accumulation assay.

The following gene sequences were used for expression in protocol 2:

*Genotype 1 UL146 (Toledo strain, 1–95)*

ACAGAACTACGATGTAGATGCTTACACAGGAAGTGGCCGCCCAACAAAATTAT
TCTGGGTAATTATTGGCTGCATCGTGATCCGCGTGGCCCAGGCTGCGACAAGAAC
GAACATCTTCTCTACCCGGACGGCCGCAAACCGCCGGGCCCGGGCGTGTGCTTGT
CGCCGGACCACTTGTTTAGCAAGTGGCTGGATAAGCATAACGATAATCGTTGGTA
TAACGTAAATATCACCAAGAGCCCGGGTCCGCGCCGTATTAACATCACCCTGATC
GGTGTCCGTGGT

*Genotype 1 UL146 (1–71)*

ACAGAACTACGATGTAGATGCTTACACAGGAAGTGGCCGCCCAACAAAATTATT
CTGGGTAATTATTGGCTGCATCGTGATCCGCGTGGCCCAGGCTGCGACAAGAACG
AACATCTTCTCTACCCGGACGGCCGCAAACCGCCGGGCCCGGGCGTGTGCTTGTC
GCCGGACCACTTGTTTAGCAAGTGGCTGGATAAGCATAACGATAATCGT

*Genotype 4 UL146 (NT strain)*

ACCGAGCTGCGTTGCAAATGCGCAGGTGGCCAAAGCTGGCATCCGAGAGGCAA
GTGGCCAACCAAACATCATTGGTTGGAGTGCTACCCGCCGAGTGGAAACTGTCCG
GCGGGCGAATTGCTGATTTATTTCGAGGAACACAACTGGAGCCCGAAGTGCGTTC
ATGTTCACAATCCGTTTGGCCAAAAATTCATGAGCAAGTGCGATAAACACGAATG
GTTCGAGGTTACCTTCAACAGCACTCGTAAGTACCCGATGATCACCCGCAAGGGC
TCGACCAAACCGACCTTCAGCTCCGGCAAA

*Genotype 5 UL146 (Davis strain)*

GGTTTTGGCACTGAGGGCAACGGCCGTTGCACGTGCATTGGCTACCACCGCCTG
TTAGGTCAACTGCCGCGGGGTACTTTCTGGCTCGGCCATCTGCCGCCAGGTAGCCA
TTGCCCGAAAGGTCAAGTTATGATCAAGATCGGTCAAGGTCCGATTGTGTGCTTGT
CCGACTACCACCCGTTGTCTAAGTGGATGTACGGCAACCACAAAAGCGGCAGCGA
AACCTGGCTGCAAATCAAAATGGAAGGTCCTAGAAATGCTACCGTGGTGCAGCG
TAGCAATACGCGTCCG

*Genotype 6 UL146 (ML1 strain)*

GGTCTGGGTAGCGAGGGGAACGGTCGTTGTACCTGTGTCGGTTATCACCGTTTT
GATAAGCAGCTGCCACGCGGTACGATCTGGCTGGGTCACCGCCCTCCGGGCCCAC
ACTGTCCGCGCGGTGACGTTCTGATGAAACTGGGTGAACAGCCGACGGTGTGTTT
GAGCGACCACCACCCACTGTCCAAATGGATGTATCGTCATCACGGCTCCGACACC
GAAATCTGGTTCCAGATTGAGTTTAAAGGTCCGCAGAATACCAAGGTGGTTTCTA
AGTCCTTTACCCCGCCGTCA

## Nuclear magnetic resonance (NMR)

NMR experiments were performed on a Bruker DRX 600 equipped with a $^1$H/$^{15}$N/$^{13}$C cryoprobe and a SampleJet robot for automated NMR screening of compounds in 96-well sample racks. NMR samples contained 90% $H_2O$, 10% $D_2O$, and 0.02% $NaN_3$, with 25 mM D13 2-ethanesulfonic acid (MES) at pH 6.8. $^1$H, $^{15}$N, and $^{13}$C resonance assignments for vCXCL1$_{GT1}$ were obtained at 25°C using the following experiments: $^{15}$N-$^1$H HSQC [63] 3D HNCA [64,65] 3D HNCO [64,66], 3D HN(CO)CA [64], 3D HNCACB, 3D HN(CA)CO, 3D HCCONH, 3D SE C(CO)NH [67], and 3D HCCH TOCSY [68]. NMR data were processed with NMRPipe [69] and XEASY [70] was used for resonance assignments and analysis of spectra. Backbone φ and ψ dihedral angle constraints and chemical shift guided predictions of secondary structure were generated from shifts of the $^1$H, $^{13}$Cα, $^{13}$Cβ, $^{13}$C′, and $^{15}$N nuclei using the program TALOS+ [31]. Heteronuclear NOE values were measured from an interleaved pair of two-dimensional $^{15}$N-$^1$H sensitivity-enhanced correlation spectra recorded with and without a 5-second proton saturation period.

## Thermal denaturation

50 μM of purified vCXCL1$_{GT1}$ and vCXCL1$_{GT1(1–71)}$ in 50 mM MES, pH 6.8 were analyzed using nanoDSF on the Nanotemper Prometheus NT.48. Protein samples and a buffer control were placed in triplicate into single use standard capillaries and heated from 20–95°C with a heating rate of 0.5°C/min. Samples were excited at 280 nm and the F350:F330 ratio was monitored during the run to determine an apparent melting temperature (Tm) for each sample.

## Inositol phosphate (IP) accumulation scintillation proximity assay (SPA)

COS-7 cells were co-transfected with receptor cDNA and the chimeric G protein Gqi4myr, which converts a $G\alpha_i$ signal into a $G\alpha_q$ readout [71,72]. One day after transfection, the cells were seeded in 96-well plates ($3.5 \times 10^4$ cells per well) and incubated with 2 μCi/ml of myo-[$^3$H]-inositol in 100 μl of growth medium for 24 h. Cells were then washed with Hanks' buffered salt solution (HBSS) and incubated at 37˚C for 15 min in 100 μl of HBSS supplemented with 10 mM LiCl. Afterwards, ligands were added and the cells were incubated for 90 min at 37˚C. When used, antagonists were added 10 min prior to the agonist. After incubation, the plates were put on ice and the medium was removed by aspiration. The cells were then lysed for 30 min by addition of 40 μl 10 mM formic acid per well, and 35 μl lysate from each well was transferred to a clean 96-well plate. Then 80 μl of a 12.5 mg/mL poly-lysine-coated yttrium silicate SPA bead suspension (Perkin-Elmer) was added to each well. The plates were sealed and incubated for 15 min at RT while shaking at medium speed followed by 5 min of centrifugation at 500 $g$. The SPA beads were allowed to react with the [$^3$H]-inositol phosphate lysate for 8 h and signal was determined on a TopCount NXT Microplate Scintillation and Luminescence Counter (Perkin Elmer). Measurements were made in duplicate.

## Bioluminescence resonance energy transfer (BRET) β-arrestin recruitment assay

HEK293 cells were seeded in 6-well plates ($6 \times 10^5$ cells per well) and co-transfected with chemokine receptor, Rluc8-arrestin3-Sp1, mem-linker-citrine-SH3, and either GRK2 or GRK6 when used. The growth medium was replaced one day after transfection. Two days after transfection, growth medium was removed by aspiration and the cells were washed with PBS supplemented with 5 mM glucose at RT. Cells were then resuspended in 3 ml PBS supplemented with 5 mM glucose per well, and 85 μl of the suspension was transferred to a 96-well black/white isoplate (Perkin Elmer) for each point. The plate was shielded from incoming light and 10 μl 50 μM coelenterazine was added to each well. After 5 min, 5 μl of ligand was added to each well and the plate was incubated at RT for 45 min in the dark. Following incubation, BRET was measured as the ratio between eYFP emission at 525 nm and Rluc emission at 485 nm using a Mithras LB 940 Multimode Microplate Reader (Berthold). Measurements were made in duplicate.

## Isolation of human peripheral blood neutrophils (PBNs)

Venous blood was drawn from a healthy adult donor, collected in EDTA tubes, and mixed by inverting the tubes a few times. The mixture was transferred to clean tube and 2% dextran sulphate in a 0.9% NaCl solution was added to the blood 1:1 (v/v), and the blood was left to sediment for 15 min at RT. All remaining steps were performed on ice/at 4˚C. After sedimentation, the upper phase was transferred to a clean tube and centrifuged for 10 min at 200 $g$, the supernatant was discarded and the pellet loosened by a quick vortexing. The pellet was resuspended in 30 ml 0.9% NaCl, vortexed, and was underlaid with 15 ml Lymphoprep (STEMCELL Technologies) and centrifuged for 30 min at 400 $g$. The supernatant was discarded, and within the next 30 sec, 10 ml sterile $H_2O$ was added while carefully mixing followed by 10 ml 1.8% NaCl, and the mixture was centrifuged for 6 min at 200 $g$. The supernatant was discarded and the pellet was visually assessed, if red, the previous lysis step was redone. If the pellet was white, it was resuspended in 5 ml 0.9% NaCl and cells were counted on a Chemometec NucleoCounter NC-3000 with the NucleoView software according to manufacturer's instructions.

## Transwell migration assay

Isolated PBNs were centrifuged for 5 min at 400 *g*, the supernatant was discarded and the cells were resuspended to a density of $2 \times 10^6$ cells/ml in sterile filtered RPMI1640 medium supplemented with 0.5% BSA (w/v). In a 3 μm pore size Corning HTS transwell 96-well plate (CLS3385, SIGMA), 220 μl chemokine ligand in RPMI1640 + 0.5% BSA was added to the bottom chamber of each transwell and 75 μl of the PMN suspension (150,000 cells/well) was added the upper chamber, mixing the cell suspension gently each time before pipetting. The plate was incubated for 1.5 h at 37˚C and 5% $CO_2$. For stably transfected L1.2 cells, 75 μl cell suspension containing 150,000 cells in RPMI1640 + 0.5% BSA medium were added per well and incubated for 4 h at 37˚C and 5% $CO_2$. After incubation, the membrane support was removed, plates were covered with a TopSeal-A PLUS (Perkin-Elmer) and centrifuged for 3 min at 300 *g*. 115 μl medium was carefully removed from each well to avoid disturbing the cell bed. The migrated cells well were resuspended by gently pipetting up and down a few times, and 100 μl was transferred to a white 96-well plate with two wells spacing between each point to avoid cross-luminescence. Light exposure was minimized and 100 μl CellTiter-Glo reagent (Promega) was added to each well. The plates were covered with TopSeal-A PLUS (Perkin-Elmer) and aluminum foil and mixed on a plate shaker for 2 min at low speed. Afterwards, the plates were incubated for 10 min at RT and luminescence was measured on an EnVision 2105 Multimode Plate Reader (Perkin-Elmer). Measurements were made in duplicate. For L1.2 cells, a standard curve was made of known cell counts and linear regression was performed to quantify the number of migrated cells.

## Rosetta comparative modeling

Rosetta comparative modeling was used to generate 5000 models for each genotype of the UL146-encoded chemokines. Representative genotype sequences corresponding to the Dolan-Davison classification [7] were acquired from the following HCMV strains and GenBank deposits: GT1 (Toledo, GU937742.2); GT2 (Merlin, AY446894.2); GT3 (KSG, AY446889.1); GT4 (NT, GQ222016); GT5 (Davis, JX512198); GT6 (ML1, AY446880); GT7 (Towne, FJ616285); GT8 (TB40, AY446866); GT9 (FS, AY446877); GT10 (AL, GQ222015); GT11 (PAV11, KJ361965.1); GT12 (6397, JX512197.1); GT13 (KM, AY446893); GT14 (RK, AY446885). Structures of CXCL2 (3N52), CXCL5 (2MGS), and CXCL8 (5WDZ) were retrieved from the Protein Data Bank (PDB) and used as templates after isolating a single chemokine chain. Target sequences were threaded onto the input templates and hybridized to construct each model, which were subsequently relaxed using the ref2015 energy function. The two disulfide bridges in canonical chemokines were defined and incorporated into model construction. Output models were evaluated using total Rosetta score and the top 10 were visually inspected. These top models were evaluated for secondary structure and similarity using root-mean-square deviation of Cα atoms. Images of protein structure were generated using PyMOL.

## Supporting information

**S1 Fig. Alignment showing the exact location of CXCL8 and vCXCL1 secondary structures.** The secondary structures of CXCL8 and vCXCL1$_{GT1-14}$ are shown in light blue (β-strands) and red (α-helices) as determined by NMR and X-ray structures for CXCL8 [22–24] and by Rosetta modelling for vCXCL1$_{GT1-GT14}$. (PDF)

**S2 Fig. Peptide mass fingerprint of vCXCL1$_{GT1}$ and vCXCL1$_{GT5}$ expressed in eukaryotic cells.** Samples of vCXCL1$_{GT1}$ **(A)** and vCXCL1$_{GT5}$ **(B)** were digested with trypsin followed by

mass spectroscopy.
(PDF)

**S3 Fig. Top models for each UL146 genotype by Rosetta. (A)** The top model (lowest Rosetta total score) is shown in an overlay for each genotype with the extended C-termini hidden for clarity. The average Cα-RMSD of core residues is 3.6 Å. All models adopt the canonical chemokine core structure. **(B)** The top models for each genotype are displayed individually. Each C-terminus forms a β-hairpin except for vCXCL1$_{GT3}$, vCXCL1$_{GT13}$, and vCXCL1$_{GT14}$.
(PDF)

**S1 Table. Comparison of HCMV encoded vCXCL1s and human chemokines with extended C-termini.** *Left side* comparison of wild-type vCXCL1s and long tailed human chemokines on secondary core structure and sequence, basic content and secondary structure of the C-terminus. *Right side* comparison of tailless wild-type vCXCL1s and tailless human chemokines with extended C-termini on secondary core structure and function.
(XLSX)

## Acknowledgments

The authors wish to thank Head of Department, MD Finn Rønholt (Department of Medicine, Herlev-Gentofte Hospital, Denmark) for support, Head of Department, Professor, dr.scient. Anders Johnsen (Department of Clinical Biochemistry, Rigshospitalet, Denmark) for his remarks on protein purification, and Olav Larsen (Laboratory for Molecular Pharmacology, Department of Biomedical Sciences, University of Copenhagen, Denmark) and Randi Mikkelsen (Novo Nordisk Foundation Center for Basic Metabolic Research, Department of Biomedical Sciences, University of Copenhagen, Denmark) for their technical assistance. This research was completed in part with computational resources and technical support provided by the Research Computing Center at the Medical College of Wisconsin.

## Author Contributions

**Conceptualization:** Hans R. Lüttichau.

**Data curation:** Christian Berg.

**Formal analysis:** Christian Berg, Michael J. Wedemeyer, Brian F. Volkman.

**Funding acquisition:** Brian F. Volkman, Mette M. Rosenkilde, Hans R. Lüttichau.

**Investigation:** Christian Berg, Michael J. Wedemeyer, Motiejus Melynis, Roman R. Schlimgen, Lasse H. Hansen, Francis C. Peterson.

**Methodology:** Christian Berg, Roman R. Schlimgen, Francis C. Peterson, Mette M. Rosenkilde, Hans R. Lüttichau.

**Project administration:** Brian F. Volkman, Mette M. Rosenkilde, Hans R. Lüttichau.

**Resources:** Brian F. Volkman, Mette M. Rosenkilde, Hans R. Lüttichau.

**Supervision:** Francis C. Peterson, Brian F. Volkman, Mette M. Rosenkilde, Hans R. Lüttichau.

**Validation:** Christian Berg, Brian F. Volkman, Mette M. Rosenkilde, Hans R. Lüttichau.

**Visualization:** Christian Berg, Michael J. Wedemeyer, Roman R. Schlimgen, Hans R. Lüttichau.

**Writing – original draft:** Christian Berg, Michael J. Wedemeyer, Roman R. Schlimgen, Hans R. Lüttichau.

**Writing – review & editing:** Christian Berg, Jon Våbenø, Brian F. Volkman, Mette M. Rosenkilde, Hans R. Lüttichau.

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
