## [Decision Letter · Decision Letter 0]

14 Jul 2021

Dear Dr. Luttichau,

Thank you very much for submitting your manuscript "The non-ELR CXC chemokine encoded by human cytomegalovirus UL146 genotype 5 contains a predicted C-terminal β-hairpin and induces neutrophil migration as a selective CXCR2 agonist" for consideration at PLOS Pathogens. As with all papers reviewed by the journal, your manuscript was reviewed by members of the editorial board and by several independent reviewers. In light of the reviews (below this email), we would like to invite the resubmission of a significantly-revised version that takes into account the reviewers' comments.

While the reviewers appreciate the quality of the work, the feel that, as presented, it is only an incremental advance over previous publications. They suggest that novelty and significance could be added by more attention to the C-terminus, and by some additional modeling.

We cannot make any decision about publication until we have seen the revised manuscript and your response to the reviewers' comments. Your revised manuscript is also likely to be sent to reviewers for further evaluation.

Sincerely,

Robert F. Kalejta

Associate Editor

PLOS Pathogens

Shou-Jiang Gao

Section Editor

PLOS Pathogens

Kasturi Haldar

Editor-in-Chief

PLOS Pathogens

orcid.org/0000-0001-5065-158X

Michael Malim

Editor-in-Chief

PLOS Pathogens

orcid.org/0000-0002-7699-2064

While the reviewers appreciate the quality of the work, the feel that, as presented, it is only an incremental advance over previous publications. They suggest that novelty and significance could be added by more attention to the C-terminus, and by some additional modeling.

Reviewer's Responses to Questions

**Part I - Summary**

Reviewer #1: The manuscript by Luttichau is a well written and comprehensive analysis of the various chemokines expressed by human cytomegalovirus (HCMV). The UL146 gene exhibits a high degree of genetic diversity based on sequencing of various clinical and laboratory virus strains. The authors provide a detailed evaluation of the sequence and structure, focusing on two variants, designated vCXCL1GT1 and vCXCL1GT2. In addition, functional analysis was performed using a variety of signaling assays, which is a strength of the study. Signaling assays included IP accumulation, beta arrestin recruitment, and cell migration. Overall, the work is soundly presented and well controlled, and the results provide an important contribution to the literature by illustrating how subtle differences in sequence and structure can be accompanied by significant changes in function and downstream signaling.

Reviewer #2: This manuscript describes the unique folding in the COOH terminus of the viral chemokines from HCMV (i.e., in the title), however the majority of the paper is about the longer/shorter versions of the amino terminus. There are interesting findings such that the non ELR motif (i.e., NGR instead of ELR) in genotype GT5 (although GT6 as well) could stimulate CXCR2 and b arrestin signaling. Also the finding that the longer amino terminus led to more activation (usually additions to the NH2 leads to less activation/antagonism such as the case with CCL5). The additional finding of this paper is that it “confirmed” the potential glycosylations via MALDI TOF.

The major flaws of this manuscripts are:

1) In order to address the role of the COOH on function there should have been experiments that truncated/swapped the different COOH termini. Instead it includes only modeling of the COOH. It’s possible that this longer COOH could function only in vivo as part of the GAG binding to establish the chemotactic gradient, which may not be detectable in vitro.

2) The novelty of this manuscript is lessened in that it is similar to the studies completed by Heo et al which used full length UL146 ORF expressed in baculovirus from 11 different genotypes (including what was called TX15 with an NGR motif). This manuscript really only addressed a single viral chemokine (GT5) while GT6 has a similar non ELR motif. This is another flaw in this manuscript. In the Heo manuscript they did not find that NRG vCXCL1 proteins activation of b arrestin or G protiens but did see some slight activation via calcium flux and upregulation of adhesion proteins. This contradicts the findings of this paper but, as the authors pointed out, it could be due to differences in the NH2 from baculovirus vs the ones tested in this paper. It would have been nice to see a comparison between TX15 from that paper and GT5 and GT6.

3) Another flaw is that this group used the bacterially expressed GT5 proteins in its functional assays (i.e., with a predetermined NH2 terminus and no glycosylations). Although their findings that the different NH2 termini length (i.e., longer functions better) is important, it would have been interesting to see whether this holds true for the other genotypes such as GT6 or the other GT with ambiguous signal cleavage sequences.

Minor points:

1) In Fig. 7 is that the GT5 (1-97)? That needs to be clarified.

2) In Fig. 7, it’s interesting that you show different bell shaped curves for the CXCR2 transfectants but similar ones with the neutrophils. That should be explored/explained.

3) Fig. 8, 9 are modeling but would be necessary to prove it with actual experiments to confirm the modeling.

Reviewer #3: Human cytomegalovirus (HCMV) encodes multiple chemokine and chemokine receptor-like genes that have been shown to be functionally active. The UL146 gene is highly diverse among sequenced clinical isolates, which suggests that this CXC chemokine is either prone to sequence evolution and/or is genetically responding to environmental pressures. Previously, the genotype GT5 (Tx15 – Heo J. Immuno 2015) has been shown to express a CXCR2 selective chemokine. The current manuscript further characterizes GT5 by determining the N’terminus by proteomics, which revealed that the full-length protein aa1-97 is the most active form, which was different than the previous report from 2015. Structural modeling predicts the extended C’terminus to encode a �-hairpin. The paper describes their results of GT5 in binding, receptor usage, GRK activation, and cellular migration. All of these assays are well done and the results are accurately described in the well written manuscript.

**Part II – Major Issues: Key Experiments Required for Acceptance**

Reviewer #1: (No Response)

Reviewer #2: (No Response)

Reviewer #3: The manuscript functionally determined and evaluated the N’terminus for GT5 identifying the most active form. The requirement and function of the C’terminal region is still unknown. The C’terminal region contains a number of basic residues which could implicate it in protein interactions. Using the assays described in this manuscript, analysis of a vCXCL1 mutant containing deletion of the C’terminal domain to mimic IL-8 or other mutations would have been informative and add to the overall impact of the manuscript. Is this region required for binding and activity or does it play an additional, yet to be defined role?

The utility of CXCR2 but not CXCR1 by UL146 GT5 is interesting and previously shown by Heo et al. In Figure 9, the authors have aligned GT1 and GT5 with IL8/CXCR2 and suggest that there are contacts R278/R208 and R212 that may allow non-ELR contacts. But what makes CXCR1 so different? Comparative modeling for CXCR1 and CXCR2 may provide insight into this unique feature especially if the structures for GT1 and GT5 are added to this analysis. Similarly, could the amino acid differences between GT1 and GT5 be used as a roadmap to identify the specific differences necessary for the effects of differential binding of CXCR1/2?

The authors pose the question about why there are 14 different known variants of UL146. In Figure 10 and in the discussion the authors provide their explanation about why HCMV encodes at least 14 different genotypes for UL146. The human sequence variation for CXCR1 and 2 is very minimal, thus host variation in CXCR1 and CXCR2 probably does not account for the driving evolutionary changes in UL146. In fact, while basic motifs are shared there is quite a bit of difference between the different genotypes of UL146. In light of this, does the suggest that the protein may be doing something else or that the minor changes are important for alterations in signaling through these receptors? A more in-depth characterization of this is really necessary to try and pick apart whether UL146 is an evolutionarily active hotspot or whether the genetic variation is/was required for viral success in the human population.

Do any of the other UL146 mutants display the same GT5 phenotype with binding CXCR2 and not 1? How about GT6? GT7 through GT14 contain a four amino acid deletion in the N’loop region, which is typically involved in receptor binding, this might also be an interesting group to assess receptor usage.

**Part III – Minor Issues: Editorial and Data Presentation Modifications**

Reviewer #1: 1. Line 60 refers to UL146 as a cytokine, whereas the title and majority of the manuscript use the term cytokine. While a chemokine is a type of cytokine, this term seems unnecessarily confusing and inconsistent here.

2. Introduction lines 74-76 – references should be provided

3. Figure 1 - clarify what black and gray shading represent

4. Fig 2 – suggest including small schematic to depict differences between two chemokine (glycosylation sites and additional 9 peptides)

5. Line 226 – “roughly half” – give exact numbers

6. Line 438 – clarify what PDB: 6LFO means

7. Line 731 – clarify cell number

Reviewer #2: (No Response)

Reviewer #3: See above

PLOS authors have the option to publish the peer review history of their article (what does this mean?). If published, this will include your full peer review and any attached files.

Reviewer #1: No

Reviewer #2: No

Reviewer #3: No
---

## [Decision Letter · Decision Letter 1]

23 Jan 2022

Dear Dr. Luttichau,

Thank you very much for submitting your manuscript "The non-ELR CXC chemokine encoded by human cytomegalovirus UL146 genotype 5 contains a C-terminal β-hairpin and induces neutrophil migration as a selective CXCR2 agonist" for consideration at PLOS Pathogens. As with all papers reviewed by the journal, your manuscript was reviewed by members of the editorial board and by several independent reviewers. The reviewers appreciated the attention to an important topic. Based on the reviews, we are likely to accept this manuscript for publication, providing that you modify the manuscript according to the review recommendations.

The reviewers found your manuscript much improved. They suggest some edits to the text, which I think are appropriate. One of the reviewers asks for some experiments that, to me, do not appear to be essential, though including them (if the data were available) would round out the manuscript. Make sure you submit a rebuttal with your responses to all the reviewers' comments, especially why you did or did not choose to include the requested experiments. Your re-submitted manuscript will undergo rapid review.

Sincerely,

Robert F. Kalejta

Associate Editor

PLOS Pathogens

Shou-Jiang Gao

Section Editor

PLOS Pathogens

Kasturi Haldar

Editor-in-Chief

PLOS Pathogens

orcid.org/0000-0001-5065-158X

Michael Malim

Editor-in-Chief

PLOS Pathogens

orcid.org/0000-0002-7699-2064

The reviewers found your manuscript much improved. They suggest some edits to the text, which I think are appropriate. One of the reviewers asks for some experiments that, to me, do not appear to be essential, though including them (if the data were available) would round out the manuscript. Make sure you submit a rebuttal with your responses to all the reviewers' comments, especially why you did or did not choose to include the requested experiments. Your re-submitted manuscript will undergo rapid review.

Reviewer Comments (if any, and for reference):

Reviewer's Responses to Questions

**Part I - Summary**

Reviewer #1: The manuscript by Berg et al describes comprehensive analysis of the UL146 isotype vCXCL1-GTA5, which lacks the classic ELR motif in the N-terminus. The authors convincingly show that vCXCL1-GTA is a elective agonist for CXCR2 only, not CXCR1 as other isotypes of vCXCL1, such as GT1, using multiple readouts that include IP accumulation, beta-arrestin recruitment, and neutrophil migration. The authors were responsive to the previous reviewer comments and the overall article is suitable for publication.

Reviewer #2: In this revised manuscript, the authors have been responsive to the many of the previous reviews by adding several new experiments, including vCXCL1tailless, a C terminal truncation, and an additional NRG motif (vCXCL1GT6). These important controls make this a much more robust study. In this process they have show another NRG vCXCL1 can stimulate CXCR2 and that with out the COOH tail, the protein miss folds. That is a very interesting finding.

**Part II – Major Issues: Key Experiments Required for Acceptance**

Reviewer #1: (No Response)

Reviewer #2: Below are some important experiments that would make this manuscript more complete:

1) The addition of MALDI data on the three new proteins in this submission vCXCL1GT4, vCXCL1GT6, and vCXCL1GT1Tailless to show the proper size and (potential) glycosylations (i.e., vCXCL1GT4 is O linked, Tailless vCXCL1GT1, vCXCL1GT5 is N linked, vCXCL1GT6 is non-glycosylated). This would expand out the Fig. 2.

2) For completion /understanding whether there are differences in b arrestin signaling (Fig. 6) and migration (Fig. 7). They show b arrestin and activation with GT1, 5, but needs GT4, 6 in Fig. 6 in Fig. 7. Who knows how well these other viral chemokines affect this signaling. Could it be biased agonism?

3) There needs to be an analysis in the discussion about their findings with tailless vCXCL1GT1 since this is a novel finding. Are there any other chemokine examples where and truncations in the C terminus affected function and/or folding?

**Part III – Minor Issues: Editorial and Data Presentation Modifications**

Reviewer #1: Key words – consider ELR motif instead of ELR

Pg 49, lines 737-742 It’s a lot of work to clone all and express all the chemokine receptors. Was this reported elsewhere that can be cited here? Also, was there a myc/His or similar tag on the receptor to verify expression? The ligand responses are convincing, but it seems like at least commenting on how receptor expression was verified would be helpful.

Pg. 61, lines 1023 and 1026 -cell number is listed as 150.000 – us a comma or scientific notation (150,000 or 1.5 x 10^5), this is confusing as written

Reviewer #2: These statements are not true:

a. Line 101 “A non-ELR CXC chemokine activates CXCR2 and adopts a unique �-hairpin

have been observed for several UL146 genotypes [10], but, apart from genotype 1, the 13

104 remaining genotypes are yet to be characterized in detail.”

This is exactly what reference 10 did in that manuscript.

b. Line 324

“Previous studies have demonstrated an effect of vCXCL1GT1 on G protein signaling, but as other pathways have not been investigated”. Again reference 10 looked at signaling of the different viral chemokines including b arrestin signaling.

c. In the discussion: Line 603: “Secondly, by searching for associations between specific viral genotypes and CMV disease, which has the potential to show the effect of the gene variants. Lastly, by performing genome-wide association studies (GWAS) of human hosts infected with HCMV to identify host and viral gene variants associated with disease development.” This statements (and Fig. 11) need to be modified to reflect the literature. UL146 variation has been addressed in both congenital infections (Bale and Demmler publications) and transplantation (Hassan Walker publication) settings so this statement should be modified. They were not able to connect any specific UL146 sequences with specific outcomes. Also, there is very little variability in human CXCR1 and CXCR2 (as mentioned by the previous reviewer) so it is not correct that viral gene variants would develop to infect different “populations” of humans more efficiently. These statements should be modified.

d. Line 331: “For CXCR2, both vCXCL1GT5 and vCXCL1GT5 stimulated β-arrestin recruitment at comparable efficacies to CXCL8 but with ~10-fold lower potencies (Fig 6D–F)” Should read both “vCXCL1GT1 and vCXCL1GT5“

PLOS authors have the option to publish the peer review history of their article (what does this mean?). If published, this will include your full peer review and any attached files.

Reviewer #1: No

Reviewer #2: No

Figure Files:

Data Requirements:

Reproducibility:

References:

---

## [Editor Report · Decision Letter 2]

9 Feb 2022

Dear Dr. Luttichau,

We are pleased to inform you that your manuscript 'The non-ELR CXC chemokine encoded by human cytomegalovirus UL146 genotype 5 contains a C-terminal β-hairpin and induces neutrophil migration as a selective CXCR2 agonist' has been provisionally accepted for publication in PLOS Pathogens.

Best regards,

Robert F. Kalejta

Associate Editor

PLOS Pathogens

Shou-Jiang Gao

Section Editor

PLOS Pathogens

Kasturi Haldar

Editor-in-Chief

PLOS Pathogens

orcid.org/0000-0001-5065-158X

Michael Malim

Editor-in-Chief

PLOS Pathogens

orcid.org/0000-0002-7699-2064
---

## [Editor Report · Acceptance letter]

4 Mar 2022

Dear Dr. Lüttichau,

We are delighted to inform you that your manuscript, "The non-ELR CXC chemokine encoded by human cytomegalovirus UL146 genotype 5 contains a C-terminal β-hairpin and induces neutrophil migration as a selective CXCR2 agonist," has been formally accepted for publication in PLOS Pathogens.

Best regards,

Kasturi Haldar

Editor-in-Chief

PLOS Pathogens

orcid.org/0000-0001-5065-158X

Michael Malim

Editor-in-Chief

PLOS Pathogens

orcid.org/0000-0002-7699-2064